# Culture In a Frame: C³B as a Comic-Based Benchmark for Multimodal Culturally Awareness

**Yuchen Song**[1,*], **Andong Chen**[1,*] **Wenxin Zhu**[1], **Kehai Chen**[2], **Xuefeng Bai**[2,†],
**Muyun Yang**[1], **Tiejun Zhao**[1,†]

[1]Harbin Institute of Technology, Harbin, China
[2]Harbin Institute of Technology, Shenzhen, China
`songyuchn@126.com, ands691119@gmail.com, wenxinzhu@stu.hit.edu.cn`
`{chenkehai, baixuefeng, yangmuyun, tjzhao}@hit.edu.cn`
🌐 : https://c3b-benchmark.github.io/

## Abstract

Cultural awareness capabilities have emerged as a critical capability for Multimodal Large Language Models (MLLMs). However, current benchmarks lack progressed difficulty in their task design and are deficient in cross-lingual tasks. Moreover, current benchmarks often use real-world images. Each real-world image typically contains one culture, making these benchmarks relatively easy for MLLMs. Based on this, we propose C³B (**C**omics **C**ross-**C**ultural **B**enchmark), a novel multicultural, multitask and multilingual cultural awareness capabilities benchmark. C³B comprises over 2000 images and over 18000 QA pairs, constructed on three tasks with progressed difficulties, from basic visual recognition to higher-level cultural conflict understanding, and finally to cultural content generation. We conducted evaluations on 11 open-source MLLMs, revealing a significant performance gap between MLLMs and human performance. The gap demonstrates that C³B poses substantial challenges for current MLLMs, encouraging future research to advance the cultural awareness capabilities of MLLMs.

## 1 Introduction

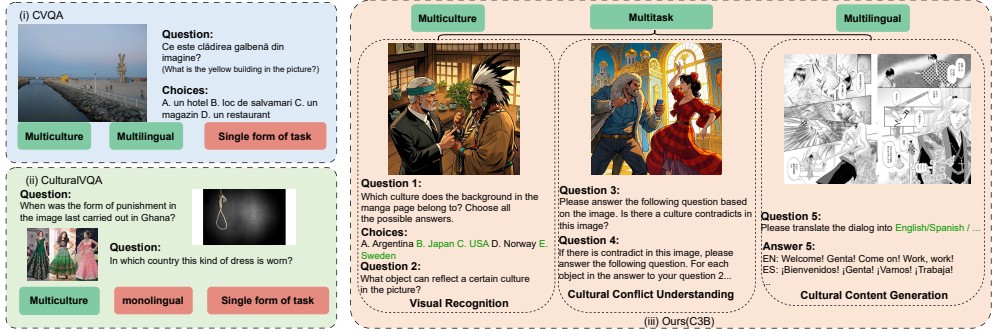

Figure 1: **Comparison between C³B and previous culture awareness capability benchmarks**. In comparison with existing benchmarks for cultural awareness capabilities, C³B is compatible with multicultural, multilingual, and multitask contexts, thereby facilitating a more thorough evaluation.

Multimodal Large Language Models (MLLMs) have become more and more important in many aspects of daily living, such as machine translation (Chen et al., 2025), image captioning (Anantha Ra-

---

*These authors contributed equally.
†Corresponding author

makrishnan et al., 2025) and visual question answering (Huynh et al., 2025). Users who interact with these models often discover such a situation: current models perform well in a Western-centric culture context but perform badly in non-Western culture contexts (Singh et al., 2025; Nayak et al., 2024; AlKhamissi et al., 2024; Burda-Lassen et al., 2025; Naous et al., 2024). This imbalance shows that MLLMs need to improve cultural awareness capabilities which refers to capabilities of MLLMs to understand and process cultural contexts (Pawar et al., 2024).

Benchmarks are essential to build MLLMs with strong cultural awareness capabilities (Cohen and Howe, 1988; Reuel et al., 2024). While existing benchmarks have laid a foundational framework for evaluation, there remains room for improvement. From a multicultral perspective, existing benchmarks for evaluating cultural awareness capabilities of MLLMs mainly focus on real-world images (Arora et al., 2025; Romero et al., 2024; Xu et al., 2025). A single real-world image typically contains one culture, making these benchmarks relatively easy for MLLMs. From a multitask perspective, most of these benchmarks include one question per data sample, which makes it difficult to evaluate MLLMs across multiple dimensions. From a multilingual perspective, languages are carriers of cultural meaning (Kramsch, 2014). A concept may have no direct equivalent expression across languages. Therefore, adding multi-lingual tasks to benchmarks will introduce appropriate complexity, enabling us to evaluate MLLMs more comprehensively.

To address these issues, we propose $C^3B$ (**C**omics **C**ross-**C**ultural **B**enchmark), a novel multicultural, multitask and multilingual cultural awareness capabilities benchmark, containing 2220 images and 18789 QA pairs. In contrast to real-world images, we select comics as the primary medium in our benchmark. Comics differ from real-world images: they often depict a fictional scene. Real-world images often tied to the specific, singular cultural contexts of real-life scenarios, but fictional scenes in comics are free from such constraints. This enables comics to condense numerous cultures into a single frame, creating a more complex context, raising the bar for evaluation. $C^3B$ consists of 3 tasks that form a logical chain. Based on difficulty levels, these tasks progress from basic visual recognition to higher-level cultural conflict understanding, and finally to cultural content generation. This multitask arrangement enables a more comprehensive evaluation of MLLMs. In cultural generation task, $C^3B$ incorporates 5 languages (Japanese, Russian, Thai, English, and Spanish), reducing the limitations of previous monolingual benchmarks. The differences between $C^3B$ and previous cultural awareness capability benchmarks are presented in Figure 1.

In this study, we conduct a comprehensive evaluation of 11 open-source MLLMs on $C^3B$. The results confirm the value of $C^3B$, as they reveal a significant performance gap between MLLMs and human performance. Furthermore, the results provide critical insights into the current state of cultural awareness capabilities of MLLMs. Specifically, MLLMs should enhance their understanding of lesser-known cultures and their abilities to process cultural conflicts. Our contributions can be summarized as follows:

1. We propose $C^3B$, a novel comic-centric, multicultural, multitask and multilingual cultural awareness capabilities benchmark.

2. $C^3B$ incorporates 3 tasks with escalating difficulty, evaluating cultural awareness capabilities of MLLMs comprehensively through progressively challenging tasks.

3. We benchmark 11 MLLMs with $C^3B$, which presents an initial set of evaluations on this benchmark, establishing a baseline for future research on MLLMs with strong cultural awareness capabilities.

## 2 RELATED WORK

### 2.1 MULTIMODAL LARGE LANGUAGE MODELS

Recent years have witnessed rapid advancements in Multimodal Large Language Models (MLLMs). The multimodal capabilities of MLLMs enable large language models to address a broader range of tasks (Caffagni et al., 2024). Models like BLIP-2 (Li et al., 2023), LLaVA (Liu et al., 2023b; 2024; 2023a) and Qwen-VL (Bai et al., 2023) perform well in tasks such as visual question answering (Dong et al., 2024) and multimodal machine translation (Chen et al., 2025). Moreover, InternLM-XC2.5 (Zhang et al., 2024) has good long-contextual input and output capabilities, which enables many advanced features, such as high resolution image understanding. Llama3.2 series

models (Grattafiori et al., 2024) are optimized for many multimodal tasks, becoming baseline for many methods.

## 2.2 Multimodal Benchmark on Cultural Awareness Capabilities

Recent multimodal benchmarks on cultural awareness capabilities mainly focus on real-world images to evaluate MLLMs. Among various recent benchmarks (Romero et al., 2024; Burda-Lassen et al., 2025; Arora et al., 2025; Schneider et al., 2025; Yang et al., 2025), CVQA (Romero et al., 2024) adopts a multiple-choice format to evaluate the cultural awareness capabilities of MLLMs using real-world images. Notably, CVQA covers 30 languages, enabling evaluation from a multilingual perspective. CVQA relies on a single form of task, restricting its ability to comprehensively evaluate the cultural awareness capabilities of MLLMs across diverse interaction scenarios. CulturalVQA (Nayak et al., 2024) also evaluates MLLMs with real-world images, covering 11 countries in 2378 images. More recently, GIMMICK (Schneider et al., 2025) incorporates six tasks for evaluating the cultural awareness capabilities of MLLMs. The images included still exhibit relatively low cultural density. Recently, GLOBALRG (Bhatia et al., 2024) has utilized images sourced from 50 countries and integrated visual grounding tasks to benchmark the cultural awareness capabilities of MLLMs. However, this framework does not account for multilingual settings. Moreover, ALM-bench (Vayani et al., 2025) encompasses over 100 languages and 19 different cultural domains, yet the images contained ALM-bench are still real-world images. AlKhamissi et al. (2025) explores the development of more sophisticated cultural benchmarks, providing a complementary lens for multimodal cultural reasoning. In parallel, Karamolegkou et al. (2024) introduces a culture-centric evaluation subset wherein a subset of images is annotated with multiple cultural associations. Overall, $C^3B$ integrates the advantages of these benchmarks, forming a multitask, multicultural, and multilingual benchmark.

## 2.3 Multimodal Benchmarks on Comics

Multimodal benchmarks centered on comics primarily focus on basic visual tasks. For Western-style comics, the eBDtheque dataset (Guérin et al., 2013) was the first publicly released comic dataset, featuring spatial and semantic annotations for 100 pages of Western comics. The COMICS dataset (Iyyer et al., 2017) includes over 1.2 million comic panels, offering resources for future research. For Japanese-style comics, Manga109 (Matsui et al., 2016) comprises 21,142 comics pages, with a primary focus on multimedia applications. More recently, CoMix (Vivoli et al., 2024) integrates comics into a new dataset but remains focused on visual tasks such as speaker identification and character naming.

Given the current landscape of multimodal benchmarks for cultural awareness capabilities and comics, we propose $C^3B$. The differences between $C^3B$ and previous works are presented in Table 1. From the table, we observe that benchmarks for evaluating cultural awareness capabilities have not simultaneously integrated multicultural images, multitask settings, and multilingual tasks. Additionally, none of them incorporate progressive difficulty tiers. Regarding comic-centered datasets, most are not designed for cultural evaluation purposes.

Table 1: **The difference between $C^3B$ and previous works.** We analyze the works along three dimensions: whether the dataset is **Multicultural** for every image, **Multitask** for every data sample, **Multilingual**, whether its primary task is **Cultural Awareness Capabilities** and whether the tasks within have **Progressed Difficulty**.

| Benchmarks | Multicultural | Multitask | Multilingual | Cultural Awareness Capabilities | Progressed Difficulty |
|---|---|---|---|---|---|
| CVQA (Romero et al., 2024) | ✗ | ✗ | ✔ | ✔ | ✗ |
| MOSAIC-1.5k (Burda-Lassen et al., 2025) | ✗ | ✗ | ✗ | ✔ | ✗ |
| CoMix (Vivoli et al., 2024) | ✔ | ✗ | ✗ | ✗ | ✗ |
| eBDtheque (Guérin et al., 2013) | ✔ | ✗ | ✗ | ✗ | ✗ |
| Manga109 (Matsui et al., 2016) | ✔ | ✗ | ✗ | ✗ | ✗ |
| CulturalVQA (Arora et al., 2025) | ✗ | ✗ | ✗ | ✔ | ✗ |
| GIMMICK (Schneider et al., 2025) | ✔ | ✔ | ✗ | ✔ | ✗ |
| $C^3B$ | ✔ | ✔ | ✔ | ✔ | ✔ |

# 3 C³B: COMICS CROSS-CULTURAL UNDERSTANDING BENCHMARK

C³B is a novel comic-centric benchmark designed to comprehensively evaluate MLLMs' cultural awareness capabilities. C³B features a multicultural diversity, a multitask setting and a multilingual coverage. To comprehensively evaluate MLLMs, C³B incorporates three tasks with increasing difficulty (detailed in Section 3.1). For data construction (detailed in Section 3.2), we design a multi-agent method to create culturally rich comics and their annotations, ensuring both efficiency and quality of images. In Section 3.3, some necessary statistics of C³B is presented. An overview of C³B is presented in Figure 2. Some data samples of C³B is presented in Appendix B.

## 3.1 TASKS

We design 3 tasks with escalating difficulty to evaluate different dimensions of cultural awareness capabilities of MLLMs. The first task, Culture-aware Object Extraction (Extraction@Culture), evaluates the visual recognition and basic cultural understanding capabilities of MLLMs. The second task, Cultural-conflict Object Detection (Conflict@Culture), focuses on evaluating their abilities to understand cultural conflicts. The third task, Culturally-aligned Content Generation (Generation@Culture), measures their multilingual generation capabilities when provided multimodal cultural contexts. The question template for each task is presented in Appendix A.

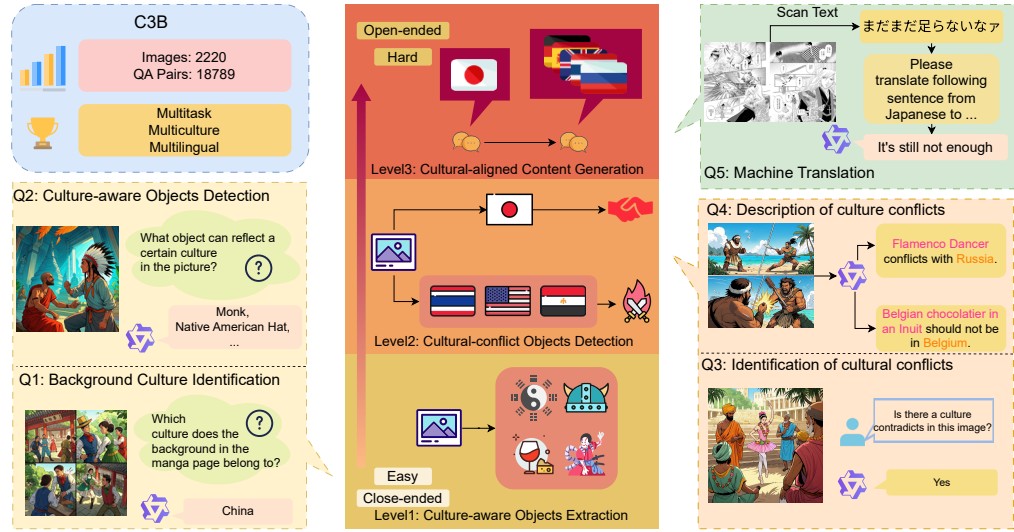

Figure 2: **Overview of C³B**. C³B evaluates MLLMs across three dimensions: Object Identification (foundational vision capability based on culture), Conflict Identification (cultural conflict understanding), and Culturally-aligned Content Generation (comprehensive cultural generation).

**Culture-aware Objects Extraction (Extraction@Culture)**: This task requires the MLLM not only to identify objects within the image but also to understand whether or not the object is related to a specific cultural context. In this task, we set two questions:

1. **Question 1 (Q1): Background Culture Identification** We task MLLMs with identifying the background culture of comic pages. Multiple valid answers may exist for a single question.

2. **Question 2 (Q2): Culture-aware Objects Detection** In this question, MLLMs are required to identify all culturally representative objects in the image. We set up multiple options, with each option including several culturally representative objects. The MLLM needs to choose an option that contains exactly all the objects present in the image.

**Cultural-conflict Objects Detection (Conflict@Culture)**: This task is designed to evaluate MLLMs' capabilities in identifying culturally conflicts within comics. In C³B, if two distinct cultures are depicted within a single image, the instance is classified as a cultural conflict. In this task, we also set two questions:

1. **Question 3 (Q3): Identification of cultural conflicts** In this question, MLLMs must determine whether there is cultural conflict in the presented image.

2. **Question 4 (Q4): Description of culture conflicts** When cultural conflict is detected, the MLLM is subsequently required to specify which objects of the answer to Q2 contradict the culture to the answer of Q1.

**Culturally-aligned Content Generation (Generaion@Culture)**: This task is designed to evaluate the cultural generation capabilities of MLLMs. As for the specific task form, we select machine translation. We have set up translation tasks for the language pairs of Japanese-English (JA-EN), Japanese-Russian (JA-RU), Japanese-German (JA-DE), Japanese-Thai (JA-TH), and Japanese-Spanish (JA-ES). These language pairs roughly cover languages from various continents, enabling a comprehensive evaluation of the overall translation capabilities of MLLMs.

## 3.2 CONSTRUCTION OF C³B

The construction process of C³B comprises two parts: image collection and annotation. An overview of the entire process is presented in Figure 3.

**Image Collection**: For task Extraction@Culture and Conflict@Culture, we design a comic generation pipeline with the help of doubao APIs[1]. The pipeline consists of two key stages: (1) prompt generation to specify cultural conflict scenarios, followed by (2) image creation based on generated prompts. This process is illustrated in Figure 3a. Subsequent to prompt generation, manual verification is performed on the generated prompts to assess the presence of harmful content.

For the Generation@Culture task, we source images from Manga109 (Matsui et al., 2016). We manually selected 1197 comics images that are closely related to culture and contain more culturally representative objects as well as cultural conflicts.

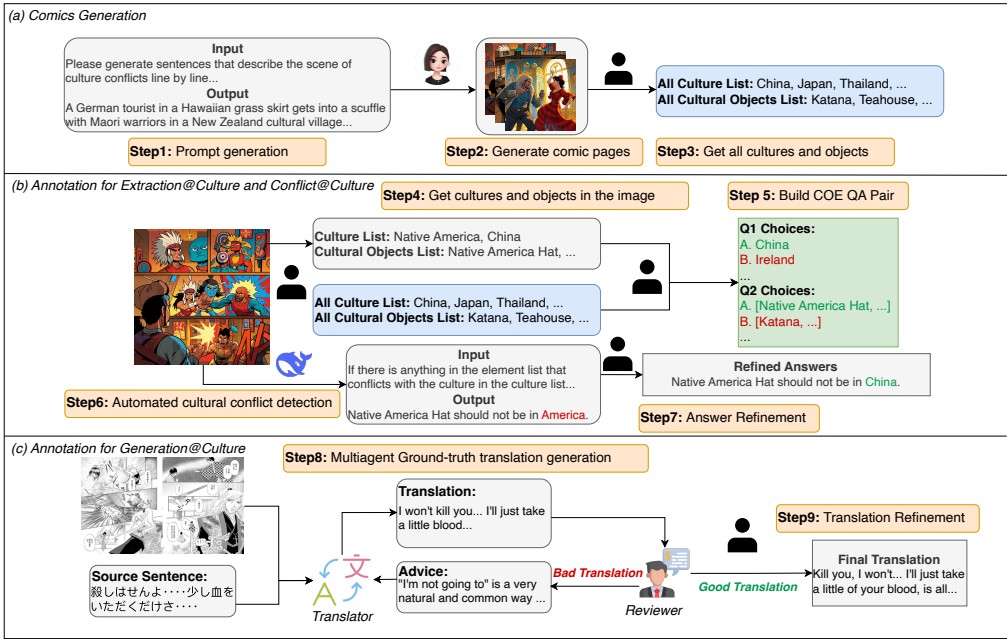

Figure 3: **The construction process of C³B**. The process contains 3 steps: Comics Generation, Annotation for Extraction@Culture and Conflict@Culture and Annotation for Generation@Culture.

**Annotation**: The annotation for the tasks of Extraction@Culture and Conflict@Culture refers to the process of QA pairs creation (illustrated in Figure 3b). For the task Extraction@Culture, we first

---

[1]https://www.volcengine.com/

manually compiled two lists: one containing all distinct cultures present in the comic pages, and the other listing all culturally representative objects. Next, we manually created QA pairs for each comic page. For Q1, we identified the cultures presented in a given image, then randomly selected additional cultures from the precompiled culture list to form a total of 5 options. For Q2, we first manually created a list of all culturally representative objects in the image, then generated 5 options by randomly modifying this list by either adding 1 object, deleting 1 object, or deleting 2 objects.

For the task Conflict@Culture, we implemented an annotation pipeline consisting of two stages with the help of Deepseek-V3 (DeepSeek-AI, 2024) (illustrated in Figure 3c). The prompt setting is presented in Appendix C.1. The details of the pipeline is:

1. **Automated Cultural Conflict Detection:** Given the background cultures and culturally representative objects in the comics page, Deepseek will analyze each object to identify if the object conflicts with one of the background cultures. After this, it generates either a formatted conflict description or "No".

2. **Manual Verification and Correction:** We manually inspect all generated results, focusing primarily on verifying whether the model misjudged the existence of conflicts. Subsequent to this, we check if the generated formatted conflict descriptions contained formatting inconsistencies or culture-related inaccuracies. The examples of these two kinds of errors are presented in Appendix D.

For the task Generation@Culture, annotation refers to the process of creating ground-truth translations. We design a multi-agent process to generate ground-truth translations, which involves two specialized agents: (1) a Translator responsible for generating translations, and (2) a Reviewer responsible for checking the translation based on the input sentence and giving suggestions. The Translator first provides a rough translation based on the extracted dialog from Manga109. If the Reviewer regards the translation as good translation, we will conduct manual verification to finalize the translation. Otherwise, the Reviewer will examine three specific types of potential errors: contextual inconsistencies, basic translation errors, and culture-related inaccuracies. The suggestions generated by the Reviewer are subsequently integrated into the prompt. The Translator then uses it to produce a new translation. In the annotation process, we employed DeepSeek-V3 as the base model, with the prompt used presented in C.2. More details on annotation are presented in E.

## 3.3 DATA STATISTICS

To ensure a comprehensive evaluation, $C^3B$ includes a total of 2,220 images and more than 18000 QA pairs. An overall statistics of $C^3B$ is presented in Table 2.

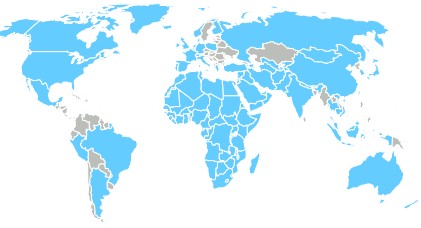

Figure 4: **The cultures $C^3B$ covers are presented in a world map.** Regions shaded in blue indicate that the culture is included in $C^3B$.

Table 2: **Statistics of $C^3B$.**

| Statistic | Number |
|---|---|
| **Images** | 2220 |
| - Extraction@Culture& Conflict@Culture | 1023 |
| - Generation@Culture | 1197 |
| - No. Japanese/Western Style Images | 1697/523 |
| **QA Pairs** | 18789 |
| - Extraction@Culture | 1023 |
| - Conflict@Culture | 1023 |
| - Generation@Culture | 16743 |
| - No. All Culture Covered | 77 |
| - Min/Max Possible Answers of Extraction@Culture | 1/5 |

In task Extraction@Culture, there are 1023 images. Each image corresponds to one QA pair. In Q1, the number of correct numbers ranges from 1 to 5 among 5 candidate options. This variability increases the difficulty for MLLMs to process the task.

In task Conflict@Culture, the images used are the same as task Extraction@Culture, covering both Japanese comics and American comic styles. This design choice is intentional because the Extraction@Culture and Conflict@Culture tasks evaluate the visual signal processing capabilities of

MLLMs. In these two tasks, introducing comics of different styles would make the process more complicated and challenging.

In task Generation@Culture, each QA pair consists of a source sentence and its ground-truth translation in 5 languages. All 16743 QA pairs cover 77 distinct cultures (visualized in Figure 4), enabling us to comprehensively benchmark the understanding different cultures of MLLMs.

# 4    $C^3B$ EVALUATION ON EXISTING LLMS

**Models**: To evaluate performance of MLLMs on our $C^3B$ benchmark, we choose 11 open-source MLLMs. Specifically, we select SPHINX (Lin et al., 2023), Monkey (Li et al., 2024), MiniGPT-v2 (Chen et al., 2023), mPLUG-Ow13 (Ye et al., 2024), LLaVA family models (Liu et al., 2023b; 2024; 2023a), InternLM-XC2.5 (Zhang et al., 2024), Llama3.2 (Grattafiori et al., 2024), Qwen2.5-VL (Bai et al., 2025) and InternVL2 (Chen et al., 2024).

**Metrics**: For task Extraction@Culture and Conflict@Culture, we use accuracy (ACC) as the evaluation metric. For Q4, because it is constructed based on models' answers to Q1 and Q2, we have designed a composite ACC metric CACC:

$$\text{CACC}(Q_4) = a \times \text{ACC}(Q_1) + b \times \text{ACC}(Q_2) + c \times \text{ACC}(Q_4) \tag{1}$$

where $a, b, c$ represent preset weighting parameters, $\text{ACC}(Q_i)$ denotes the accuracy score for Question $i$. These weighting parameters quantify the respective contributions to the final evaluation. In our experiments, the values of $a, b, c$ are 0.3, 0.3 and 0.4. Q1 and Q2 contribute approximately equally to Q4, while Q4 should dominate. To simultaneously consider the contributions of the three tasks, we set the hyperparameters in this way.

For Generation@Culture, we adopt the conventional metric BLEU (Papineni et al., 2002), supplemented by COMET (Rei et al., 2022) and BLEURT (Sellam et al., 2020) to align with current standards in LLM-based translation research.

**Evaluation Settings** All evaluations were conducted on a Ubuntu server equipped with an H20-NVLink GPU featuring 96GB of memory. We adhered strictly to the official inference example codes provided by the respective model developers.

# 5    EXPERIMENT RESULTS

## 5.1    MAIN RESULTS FOR TASK EXTRACTION@CULTURE AND CONFLICT@CULTURE

Table 3: **Main Results of task Extraction@Culture and Conflict@Culture in $C^3B$**. For task Conflict@Culture, two types of accuracy metrics are used to evaluate model performance. Specifically, Q4 is constructed based on the answer of MLLM to Q1 and Q2. The calculation process of **CACC** of Q4 considers the answers to Q1 and Q2. In contrast, **ACC** refers to the accuracy of Q4 calculated without referencing answers to Q1 and Q2. The highest performance is marked in **bold**.

| Methods | Extraction@Culture | | Conflict@Culture | Conflict@Culture (Q4) | |
|---|---|---|---|---|---|
| | Q1 (ACC) | Q2 (ACC) | Q3 (ACC) | ACC | CACC |
| **SPHINX** | 29.9 | 5.28 | **69.2** | 0.16 | 10.6 |
| **Monkey** | 28.4 | 5.08 | 33.0 | 0.19 | 10.1 |
| **MiniGPT-v2** | 18.0 | 6.84 | 66.3 | 0.51 | 7.7 |
| **mPLUG-Owl3** | 24.9 | 15.2 | 30.8 | 2.03 | 12.8 |
| **LLaVA1.5-7B** | 32.5 | 2.93 | 56.3 | 0.00 | 10.6 |
| **LLaVA-NeXT** | 16.5 | 39.8 | 0.88 | 0.00 | 16.9 |
| **LLaVA-OV** | 39.0 | 32.4 | 53.1 | **3.87** | 23.0 |
| **InternLM-XC2.5** | 46.0 | 50.9 | 68.5 | 1.94 | 29.8 |
| **Llama3.2** | 46.0 | **59.0** | 44.9 | 2.76 | 32.6 |
| **Qwen2.5-VL** | **53.7** | 55.9 | 63.1 | 3.20 | **34.2** |
| **InternVL2** | 46.0 | 50.9 | 68.5 | 0.01 | 29.1 |

Table 3 presents the results for task Extraction@Culture and Conflict@Culture. Qwen2.5-VL demonstrates optimal performance, with Q1 outperforming the second-ranked models (InternLM-

XC2.5, Llama3.2 and InternVL2) by 7.7 points and Q4 achieving a 1.6-point lead over the second-ranked model (Llama3.2). We also find that all models achieve extremely low **ACC** in Q4 due to the influence from Q1 and Q2.

As for task Extraction@Culture, in Q1, LLaVA-NeXT performs the worst, because it tends to describe the image rather than answer the question directly. We name this phenomenon as "Turn-a-deaf-ear" (shown in Appendix F.1). In Q2, Llama3.2 performs the best, exceeding average performance by 72.7%. In this task, LLaVA1.5-7B performs the worst, because it keeps answering "A". We name this failure pattern as "Take-a-shot-in-the-dark" and an example is shown in Appendix F.2.

As for task Conflict@Culture, in Q3, SPHINX performs the best, exceeding the second-ranked model (InternLM-XC2.5 and InternVL2) by 0.7 points. In Q4, we find that models from the LLaVA series yield considerably lower results. Specifically, both LLaVA1.5-7B and LLaVA-NeXT achieves 0.00 ACC. Through case study (detailed in Appendix F.3), we find that, LLaVA-NeXT persistently outputs "Nothing", indicating a lack of cultural conflict comprehension capability, while LLaVA1.5-7B cannot follow instructions properly, and we name this phenomenon as "stubbornness".

## 5.2 MAIN RESULTS FOR TASK GENERATION@CULTURE

Table 4 shows the results for task Generation@Culture. The results show that Qwen-2.5-VL demonstrates the most robust multilingual capabilities, as it outperforms other models across all cultural generation tasks. In contrast, MiniGPT-v2 exhibits notably poor performance. Specifically, it achieves a BLEU score of 0 in most tasks. This underperformance can be attributed to the weak instruction-following ability, as presented in Appendix F.4. This error pattern might be caused by the poor understanding capability of comic pages. Additionally, LLaVA-NeXT tends to repeat the source sentence when handling the tasks except JA-EN. Among all tasks, the performance of all models in JA-TH is the poorest while they all perform the best in JA-EN tasks. These results indicate that the multilingual capabilities when given cultural contexts needs to be enhanced, and greater support for low-resource languages should also be strengthened.

Table 4: **Main Result of task Generation@Culture**. The result is presented in the format BLEU/COMET/BLEURT. The highest performance is marked in **bold**.

| Methods | JA-EN | JA-DE | JA-RU | JA-ES | JA-TH |
|---|---|---|---|---|---|
| **SPHINX** | 3.71/63.2/42.3 | 1.12/54.3/33.1 | 0.27/45.0/19.3 | 1.62/56.9/27.5 | 0.02/38.7/16.1 |
| **Monkey** | 3.88/62.0/43.3 | 1.62/55.5/33.5 | 0.73/48.2/17.7 | 2.81/56.3/29.3 | 0.63/50.2/18.0 |
| **MiniGPT-v2** | 0.03/44.5/30.9 | 0.00/32.1/15.8 | 0.00/30.0/13.9 | 0.00/36.5/21.6 | 0.00/32.2/14.5 |
| **mPLUG-Owl3** | 6.21/58.4/39.8 | 4.97/56.0/36.8 | 3.97/57.6/33.5 | 4.75/55.7/34.6 | 2.02/51.9/25.2 |
| **LLaVA1.5-7B** | 5.94/62.0/41.0 | 2.74/54.2/29.7 | 1.39/53.8/23.1 | 4.23/62.2/38.4 | 1.13/44.9/10.2 |
| **LLaVA-NeXT** | 4.88/61.2/41.2 | 0.00/46.4/10.5 | 0.14/41.8/11.1 | 0.00/48.7/6.87 | 0.28/42.1/12.5 |
| **LLaVA-OV** | 6.00/58.3/35.2 | 3.37/49.6/24.7 | 2.78/49.3/23.0 | 3.24/53.4/26.5 | 3.77/42.8/13.8 |
| **InternLM-XC2.5** | 4.99/66.6/49.9 | 1.08/59.2/42.0 | 1.90/63.4/40.6 | 2.33/65.3/44.7 | 0.53/53.7/27.3 |
| **Llama3.2** | 5.05/54.6/35.2 | 4.27/51.3/31.8 | 1.70/47.0/20.1 | 5.73/55.0/31.1 | 0.99/46.9/17.4 |
| **Qwen2.5-VL** | **13.2/70.9/53.8** | **12.0/67.7/52.1** | **8.74/69.8/48.5** | **14.5/72.0/54.3** | **9.72/67.8/42.2** |
| **InternVL2** | 7.20/66.6/47.6 | 3.52/58.5/40.4 | 3.33/63.8/40.3 | 5.32/66.8/46.9 | 0.74/45.3/24.5 |

## 5.3 IMPACT FOR Q4 FROM Q1 AND Q2

In $C^3B$, answering Q4 requires models to correctly answer Q1 and Q2. To evaluate the influence of Q1 and Q2 on Q4, we first calculate the correlation coefficients (Formula 2) between Q1 and Q4 and between Q2 and Q4, which yielded values of 0.56 and 0.51, respectively. The result demonstrates that Q1 and Q2 exhibit a moderate correlations with Q4.

$$R(Q_i, Q_4) = \frac{\text{Cov}(\text{ACC}(Q_i), \text{ACC}(Q_4))}{\sqrt{\text{Var}(\text{ACC}(Q_i))\text{Var}(\text{ACC}(Q_4))}} \tag{2}$$

where $R$ represents the correlation coefficient between two questions, Cov refers to the covariance, Var refers to the variance, and $\text{ACC}(Q_i)$ is the accuracy value of Question $i$.

Moreover, we conducted an ablation study on Q1 and Q2. We added the answers to Q1/Q2 into the prompt template of Q4 to observe the impact on model performance. **CACC** represents the average performance of all 11 MLLMs, and the results are presented in Table 5. The results indicate that incorporating the Q1 answer into the Q4 prompt will enhance performance (+0.003), but adding the Q2 answer will not change the performance. The greatest performance boost is observed when both Q1 and Q2 answers are included (+0.533). Although Q1 and Q2 show moderate correlations with Q4 (0.56 and 0.51), the ablation results reveal only marginal improvement when adding Q1/Q2 answers. This suggests that the positive correlations mainly reflect an intrinsic consistency across related questions, while the direct prompt incorporation of Q1/Q2 answers is not always effectively leveraged by MLLMs.

Table 5: **Performance of Q4 when answers to Q1 and/or Q2 are omitted.** Q1/Q2 Answer means that only Q1/Q2 answer is provided. Q1&Q2 Answer means both Q1 and Q2 answers are provided.

| Method | CACC |
|---|---|
| Base Prompt | 19.231 |
| + Q1 Answer | 19.234 |
| + Q2 Answer | 19.231 |
| + Q1&Q2 Answer | 19.764 |

### 5.4 GENERATED COMIC PAGES ANALYSIS

To evaluate C³B's cultural diversity, we compute three measuresCulture Density Per Image (CDPI), Cultural Breadth Intensity (CBI), and Coverage-Adjusted Density (CAD)and compare them against cultural QA datasets. CDPI is defined as the quantification of the average number of distinct cultures depicted within an image. CBI is designed to measure both cultural density and the total number of cultures depicted in an image. CAD incorporates log-scaling to alleviate excessive rewards arising from large cultural lists. The CDPI, CBI and CAD of a dataset are presented in Equation 3, 4 and 5. In the equations $D$ represents a complete image dataset, which is $\{I_1, I_2, \cdots, I_n\}$, CultureInImage($I_i$) denotes the number of cultures in the image $I_i$, $N_{\text{cultures}}$ refers to the number of cultures occurring in the dataset and $|D|$ as the cardinality of $D$.

$$\text{CDPI}(D) = \frac{1}{|D|} \sum_{i=1}^{n} \text{CultureInImage}(I_i) \tag{3}$$

$$\text{CBI}(D) = \text{CDPI}(D) \times N_{\text{cultures}} \tag{4}$$

$$\text{CAD}(D) = \text{CDPI}(D) \times \log_2(N_{\text{cultures}} + 1) \tag{5}$$

The results are presented in Table 6, indicating that C³B shows significantly higher cultural diversity compared to similar datasets. This highlights its improved ability to evaluate cultural awareness capabilities of MLLMs.

Table 6: **Statistics of different culture-related datasets**.

| Dataset | CDPI | CBI | CAD |
|---|---|---|---|
| CVQA | 1.00 | 30 | 4.91 |
| CulturalVQA | 1.00 | 11 | 3.58 |
| MOSAIC-1.5k | 1.00 | 43 | 5.46 |
| GIMMICK | 1.00 | 144 | 7.18 |
| **C³B** | **2.28** | **175.56** | **14.29** |

### 5.5 SCORES OF DIFFERENT CULTURE

To examine how well MLLMs can recognize diverse cultures, we conducted a culture-specific evaluation of models' correctness to mono-culture QA task (Q1). The results, visualized in Figure 5, indicate that the performance of MLLMs differs substantially across cultural groups. In detail, representative cultures (e.g., Cambodia and Japan) are reliably identified, while lesser-known cultures (e.g., Finnish and Somalia) exhibit notably higher error rates. The results show that the cultural awareness capabilities of MLLMs for lesser-known cultures should be enhanced.

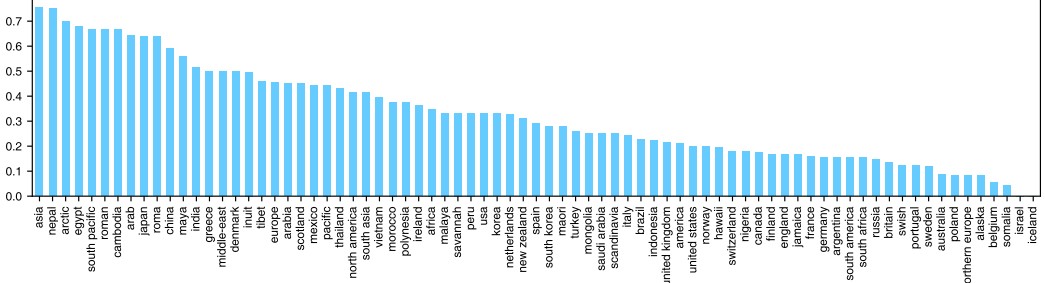

Figure 5: **Model Accuracy on Q1 by Culture.**

## 5.6 HUMAN-MODEL ANALYSIS

To validate the effectiveness of C³B, we conducted a human evaluation (more detail in Appendix G). First, We categorized task difficulties based on the number of distinct cultures in each image: 1 (easy), 2/3 (medium), and 4/5 (hard). This tiered classification allows evaluation across different complexity levels. Next, we randomly selected 100 questions per tier. For each tier, we calculated two metrics: the average accuracy performance of human and MLLMs. The results are presented in Table 7. Three graduate student volunteers participated in the experiment, with each completing all the questions. We then calculated their average accuracy to quantify their performance.

Table 7: **Result of Human-Model Analysis**. Q4 is constructed based on the answer of MLLM to Q1 and Q2. The calculation process of ACC of Q4 considers the answers to Q1 and Q2. In contrast, **ACC** refers to the accuracy of Q4 calculated without referencing models' answers to Q1 and Q2. **IRA** stands for Inter-rater Agreement Score. **PTA** stands for Per-tier Time-to-answer.

| Method | Q1 | Q2 | Q3 | Q4 | | IRA | PTA(Seconds) |
|---|---|---|---|---|---|---|---|
| | ACC | ACC | ACC | ACC | CACC | | |
| *Easy Questions* | | | | | | | |
| **Human Evaluation** | 92.0 | 77.0 | 100.0 | 60.0 | 74.7 | 95% | 27.01 |
| **Model Performance** | 48.0 | 28.8 | 50.5 | 1.90 | 23.8 | - | - |
| *Medium Questions* | | | | | | | |
| **Human Evaluation** | 78.0 | 60.0 | 100.0 | 45.0 | 59.4 | 91% | 52.92 |
| **Model Performance** | 32.6 | 27.1 | 54.0 | 1.69 | 18.6 | - | - |
| *Hard Questions* | | | | | | | |
| **Human Evaluation** | 69.0 | 51.0 | 100.0 | 35.0 | 50.0 | 93% | 60.61 |
| **Model Performance** | 25.7 | 25.5 | 41.7 | 1.27 | 15.9 | - | - |

We find that the human performance is significantly better than that of MLLMs, particularly in Q3, where the human performance achieves all 100% accuracy. Constrained by the performance in Q1 and Q2, the human ACC result is relatively low, yet it still exceeds the MLLMs' performance. The results show that C³B is challenging for MLLMs and the cultural awareness capabilities of MLLMs need to be improved.

## 6 CONCLUSION

We propose C³B, a novel comic-based, multicultural, multitask, and multilingual cultural awareness capabilities benchmark. Applying comics as the core medium, we enable comprehensive evaluation of cultural awareness capabilities of MLLMs, with the dataset encompassing a large number of images, QA pairs, and cultural contexts. C³B contains three tasks with progressed difficulties, from basic visual recognition to higher-level cultural conflict understanding, and finally to cultural content generation. We benchmarked 11 open-source MLLMs on C³B, revealing a significant performance gap between MLLMs and human performance. Specifically, current MLLMs lack proficiency in understanding less well-known cultures and processing cultural conflicts, highlighting areas for improvement. We anticipate that C³B will serve as a critical tool to support and advance research on MLLMs' cultural awareness capabilities.

## 7 ACKNOWLEDGEMENTS

We want to thank all the anonymous reviewers for their valuable comments. The work was supported by the National Natural Science Foundation of China (62376075, 62276077, 62406091, 62350710797 and U23B2055), Guangdong Basic and Applied Basic Research Foundation (2024A1515011205, 2026A1515011718) and Shenzhen Science and Technology Program (KQTD20240729102154066 and ZDSYS20230626091203008).

## ETHICS STATEMENT

Our C³B benchmark incorporates images created by Doubao. Through a rigorous human annotation and verification process (Appendix E), we have made every effort to mitigate the majority of cultural biases and stereotypes. Prior to the creation of images, we manually checked all prompts and removed harmful content, thereby ensuring the high quality of the created images.

Regarding the definition of cultural conflict, we have explicitly clarified that the cultural conflict in our task refers to co-occurrence conflict rather than aggressive conflict. Our work is not intended to cause any harm to individuals or groups.

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

**Input Image**:

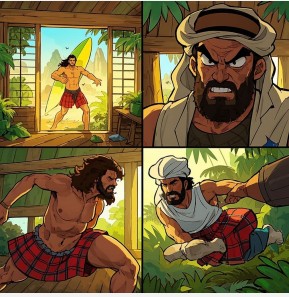

**Q1**: Which culture does the background in the comics page belong to? Choose all the possible answers.
**Choices for Q1**:
A. Cambodia
B. Greece
C. Brazil
D. Australia
E. China
**Answer to Q1**: C D
**Q2**: What object can reect a certain culture in the picture? You should choose the most appropriate answer.
**Choices for Q2**:
A. ['An Australian surfer', ' kilt', ' Arabian sheikhs in a Brazilian', ' rainforest hut', 'indian sari']
B. ['An Australian surfer', ' kilt', ' Arabian sheikhs in a Brazilian', ' rainforest hut', 'batik robe has a clash']
C. ['An Australian surfer', ' kilt', ' Arabian sheikhs in a Brazilian', ' rainforest hut', 'a group of inuit elders']
D. ['An Australian surfer', ' kilt', ' Arabian sheikhs in a Brazilian', ' rainforest hut']
E. ['An Australian surfer', ' kilt', ' Arabian sheikhs in a Brazilian']
**Answer to Q2**: D
**Q3**: lease answer the following question based on the image. Is there a culture contradicts in this image?
**Answer to Q3**: Yes.
**Q4**: If there is contradict in this image, please answer the following question. For each object in ['An Australian surfer', ' kilt', ' Arabian sheikhs in a Brazilian', ' rainforest hut'], if that object and a certain culture in ["Brazil", "Australia"] conflict with each other, please output it. The format is "Something should not be in Some Culture". For example, "Katana should not be in America", "Chinese calligrapher should not be in Africa". If there isn't contradict, output nothing.
**Answer to Q4**:
1. An Australian surfer should not be in Brazil.
2. kilt should not be in Australia.
3. kilt should not be in Brazil.
4. Arabian sheikhs in a Brazilian should not be in Australia.
5. rainforest hut should not be in Australia.

Figure 6: **A data sample of Extraction@Culture and Conflict@Culture.**

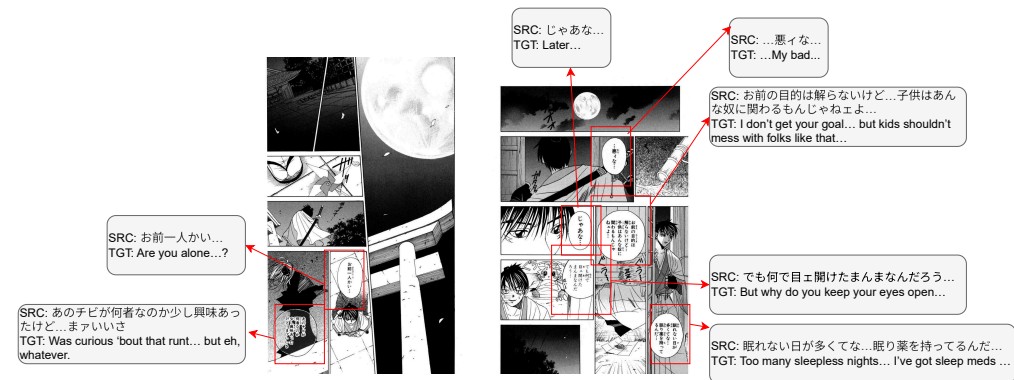

Figure 7: **A data sample of Generation@Culture**. During each inference, a single source sentence and the comic page are input to the MLLM.

## A  QUESTION SETTING FOR EACH TASK

### A.1  EXTRACTION@CULTURE

Question 1: Which culture does the background in the comics page belong to? Choose all the possible answers.
Question 2: What object can reflect a certain culture in the picture? You should choose the most appropriate answer.

### A.2  CONFLICT@CULTURE

Question 3: Please answer the following question based on the image. Is there a culture contradicts in this image?
Question 4: If there is contradict in this image, please answer the following question. For each object in the $A_2$, if that object and a certain culture in $A_1$ conflict with each other, please output it. The format is "Something should not be in Some Culture". For example, "Katana should not be in America", "Chinese calligrapher should not be in Africa". If there isn't contradict, output nothing.

where $A_1$ refers to the list of cultures of the MLLM's answer to Q1, and $A_2$ refers to the list of cultural representative objects of the MLLM's answer to Q2.

### A.3  GENERATION@CULTURE

Translate following sentences from Japanese into English. Output the result line by line. The sentences are:

## B  SOME DATA SAMPLE FROM C$^3$B

### B.1  EXTRACTION@CULTURE AND CONFLICT@CULTURE

An example of Extraction@Culture and Conflict@Culture is presented in Figure 6.

## B.2 Generation@Culture

An example of Generation@Culture is presented in Figure 7.

# C Prompt Setting for Data Collection and Annotation

## C.1 Prompt Setting for QA Pair Creation of Conflict@Culture Task

> Only output the answer. You need to check if there is anything in the element list that conflicts with the culture in the culture list. If there is, you should output the result as "Something should not be in Somewhere.". Otherwise, just output "No".
> For an element, if it conflicts with any one of the culture in the list, you should count it as aconflict. For example, the culture list is [Japan, China] and the element list is [katana, Chinese tea]. Then you should output "katana should not be in China" and "Chinese tea should bein Japan".
> As for acceptable results. For example, some acceptable results are "Katana should not be inAmerica", "Chinese calligrapher should not be in Africa" and "No".
> Now output the result for: The element list is $l_e$. The culture list is $l_p$.

Figure 8: **The prompt setting for QA pair creation in Conflict@Culture Task**.

In the prompt setting of Conflict@Culture (presented in Figure 8), $l_e$ and $l_p$ denote the list of culturally representative objects and culture of image annotated in task Extraction@Culture respectively.

## C.2 Prompt Setting for Annotation in Generation@Culture

The model we apply in annotation in Generation@Culture for creating C$^3$B is DeepSeek-R1 (DeepSeek-AI, 2025). The prompt we use for Translator is presented in Figure 9. The prompt we use for Reviewer is presented in Figure 10. Upon receiving the review feedback, the prompt provided to the Translator is presented in Figure 11.

> Output the answer line by line. Please only output the answer. Translate following sentence from Japanese to English: $s$.

Figure 9: **The prompt used for the Translator**, where $s$ denotes the source sentence that is to be translated.

> Output the answer line by line. Only output the translation, not suggestions. Please only output the modified translation. You need to pay attention to whether there are any inconsistencies in context, whether there are culturally related errors and whether there are any translation errors. The source is:$s$ and the translation is:$t$.

Figure 10: **The prompt used for the Reviewer**, where $t$ denotes the translation provided by the Translator.

> Output the answer line by line. Please only output the answer. Translate following sentence from Japanese to English: $s$. Here is some advice: $a$

Figure 11: **The prompt used for the Translator once the review is provided**, where $a$ denotes the review generated by the Reviewer.

> **Input**: Only output the answer. You need to check if there is anything in the element list that conflicts with the culture in the culture list. If there is, you should output the result as "Something should not be in Somewhere.". Otherwise, just output "No".
> For an element, if it conflicts with any one of the culture in the list, you should count it as a conflict. For example, the culture list is [Japan, China] and the element list is [katana, Chinese tea]. Then you should output "katana should not be in China" and "Chinese tea should not be in Japan".
> As for acceptable results. For example, some acceptable results are "Katana should not be in America", "Chinese calligrapher should not be in Africa" and "No".
> Now output the result for: The element list is [China, Japan, Brazil]. The culture list is [Chinese calligrapher, Japanese Katana, Brazilian Dancer].
> **Output with Error**: In China, Brazilian dancers are rare...
> **Corrected Output**: Brazilian dancers should not be in China.

Figure 12: **An example of formatting inconsistency.** The format provided is shaded in blue, and the output with error doens't follow it.

## D    EXAMPLES OF ERRORS IN MANUAL VERIFICATION IN CONFLICT@CULTURE

### D.1    FORMATTING INCONSISTENCY

This type of error occurs when the format of the conflict descriptions generated by the models fails to conform to the format provided. An example is provided in Figure 12.

### D.2    CULTURE-RELATED ERRORS

This type of error occurs when the conflict descriptions generated by the models contains cultural errors. An example is provided in Figure 13.

## E    ANNOTATION DETAILS

All annotations were performed by three graduate students and two undergraduate students, with distinct division of labor across annotation tasks. All of the annotators come from Asian countries. Among the five annotators, three are graduate students engaged in culture-related research. The quality of annotation is ensured by credible online resources (e.g. Wikipedia).

For the annotation of cultural lists and cultural object lists in Image Collection phase, the two undergraduate students were tasked with this work, where each was assigned 512 images (accounting for 50% of the total dataset) for annotation. Subsequent to this initial annotation phase, the three graduate students conducted a comprehensive review of the annotated content and rectified any annotation errors identified based on the prompts which are used to generate images. The criteria for this phase are defined as follows:

1. **Fidelity**: An object is counted if it evokes a perceived association with a specific cultural context.

> **Input**: Only output the answer. You need to check if there is anything in the element list that conflicts with the culture in the culture list. If there is, you should output the result as "Something should not be in Somewhere.". Otherwise, just output "No".
>
> For an element, if it conflicts with any one of the culture in the list, you should count it as a conflict. For example, the culture list is [Japan, China] and the element list is [katana, Chinese tea]. Then you should output "katana should not be in China" and "Chinese tea should not be in Japan".
>
> As for acceptable results. For example, some acceptable results are "Katana should not be in America", "Chinese calligrapher should not be in Africa" and "No".
>
> Now output the result for: The element list is [China, Japan, Brazil]. The culture list is [Chinese calligrapher, Japanese Katana, Brazilian Dancer].
>
> **Output with Error**: Japanese Katana should not be in Japan.
>
> **Corrected Output**: Japanese Katana should not be in China. Japanese Katana should not be in Brazil.

Figure 13: **An example of culture-related errors.** In our example, Japanese katana should not be in the country except Japan.

2. **Granularity**: For an object associated with a culture shared across multiple countries, a regional-level description shall be adopted.

3. **Security**: An image shall be excluded if it contains aggressive depictions, which refer to racist content, sexual explicitness, and other analogous content that may cause harm to individuals.

Following the annotation rectification phase, we calculated that 237 out of the total 1,023 images underwent revisions. The inter-annotator agreement score was determined to be 76.8%.

For the annotation of question-answer (QA) pairs corresponding to the three tasks, all five annotators participated in the verification phase. Initially, each annotator independently revised the annotation results; subsequent to this individual revision step, a joint review of each question was conducted collectively by the entire annotation team. Finally, the questions corresponding to 755 images were manually revised, resulting in a correction rate of 73.8%. In this phase, the criteria are defined as follows:

1. **Culture Fidelity**: All annotators were instructed to retrieve relevant information from credible online sources (e.g. Wikipedia) to verify the accuracy of the specific forms of cultural objects depicted in the image. In cases where inaccuracies were identified, the annotations were revised based on the outcomes of group discussions.

The guidelines for the annotation of cultural lists and cultural object lists are presented as follows:

1. **Fidelity & Granularity**: If an element in the image shows a country's cultural traits, please label it with the country name. If it reflects a regional cultural feature, please use the region or continent namelike "Middle East," "Asia," "South Pacific," and so on.

2. **Security**: If there's any stereotype or offensive content in the image or prompt, mark it for removal.

3. **Example**: When labeling elements in the image, try to keep the labels as close as possible to the prompt used to generate the image. Here's an example: "A Brazilian footballer in a Scottish kilt has a shouting match with Chinese ping-pong players in a Beijing sports complex." In such case, if the elements mentioned in the prompt appear in the image and can be matched to the entities in the prompt (e.g., the footballer in the image corresponds to a "Brazilian footballer," etc.), label them exactly as "Brazilian Footballer," "Scottish Kilt," and so on.

4. **Credibility**: For the cultural elements in the image, look them up on Wikipedia if you are not familiar with the culture. You can label them only if they match the relevant information there; otherwise, skip the labeling.

# F CASE STUDY

During evaluation, we observed several unexpected behavioral patterns that led to suboptimal question-answering performance in MLLMs.

## F.1 ERROR CASES IN Q1

**"Turn-a-deaf-ear"**: This behavioral pattern is particularly occurred in LLaVA-NeXT, where the model frequently defaulted to describing the image rather than direct question-answering when presented with clear instructions. We name this behavoir as "Turn-a-deaf-ear". An example is provided in Figure 14.

We assume that this behavioral pattern likely stems from the fine-tuning dataset's compositional bias, which predominantly trains MLLMs for image description tasks rather than image-grounded question answering.

We need to mention that, in Q3, LLaVA-NeXT fails to predict the result properly due to the same reason as in Q1.

---

**Input Image**:

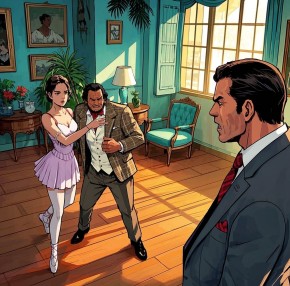

**Input**: Which culture does the background in the comics page belong to? Choose all the possible answers. Please only generate answer.
A. Australia
B. France
C. Middle-east
D. Indonesia
E. South america
**Ground-truth Answer**: A,B,C
**Output**: The background in the comics page appears to be inspired by a setting that could be from a variety.

---

Figure 14: **An example of "Turn-a-deaf-ear" is presented.** The MLLM is asked to output the correct choices but it outputs the detail of the comics page instead.

## F.2 ERROR CASE IN Q2

**"Take-a-shot-in-the-dark"**: This behavioral pattern was particularly observed in LLaVA1.5-7B when the model attempted to answer Q2. For this question, the model exhibited a tendency to output "A" as the response.

We calculated the frequency that LLaVA1.5-7B outputs the choice "A" and the result is 78.4, which says that the model cannot understand the cultures properly and tends to output an answer.

### F.3 ERROR CASE IN Q4

**Keep answering "Nothing"**: This behavioral pattern is particularly occurred in LLaVA-NeXT, where the model always answering "Nothing", which shows a lack of cultural conflict comprehension capability. An example is presented in Figure 15.

**Input Image**:

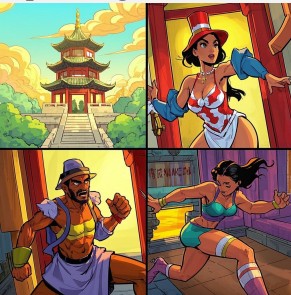

**Input**: If there is contradict in this image, please answer the following question. For each object in ['Brazilian samba dancer', ' Swiss dirndl engages', ' dispute with Turkish Janissaries', ' Chinese pagoda'], if that object and a certain culture in the [Turkey, Switzeland, Brazil, China] conflict with each other, please output it. The format is "Something should not be in Some Culture". For example, "Katana should not be in America", "Chinese calligrapher should not be in Africa". If there isn't contradict, output nothing.
**Ground-truth Answer**: "Swiss dirndl engages should not be in Brazil, · · · "
**Output**: Nothing.

Figure 15: **An example of keeping answering nothing**. MLLM keeps answering "nothing" instead of the description of cultural conflicts.

**"Stubbornness"**: This behavioral pattern is particularly occurred in LLaVA-1.5-7B, where the model frequently defaulted to follow the instruction without thinking. We name this behavior as "Stubbornness". An example is provided in Figure 16.

### F.4 ERROR CASE IN Q5

**Poor instruction following capabilities of MiniGPT**: For Question 5, our instruction to MLLMs was to directly output the translated sentence without including any redundant explanatory content. Nevertheless, MiniGPT tended to elaborate on how the translation was performed and also repeated the identical sentence a second time. An example is presented in Figure 17.

## G DETAILS ON HUMAN-MODEL ANALYSIS

Regarding the human-model analysis, three undergraduate students volunteered to participate. Each question was answered separately, with a one-hour time limit per question, and was presented in the same format used for the MLLMs.

## H EFFECT OF OCR INTEGRATION ON MT TASK PERFORMANCE

We conducted additional experiments to evaluate the impact of OCR on the Generation@Culture task. Our experimental settings are as follows: one group directly adopted the pre-extracted text from Manga109, while the other group utilized OCR[2] technology for text extraction. We use BLEU as the evaluation metric The results are presented in Table 8.

---

[2]https://github.com/PaddlePaddle/PaddleOCR

**Input Image**:

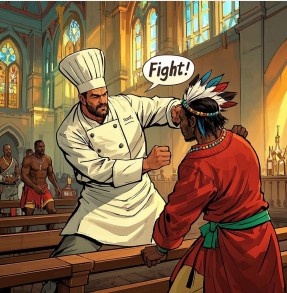

**Input**: If there is contradict in this image, please answer the following question. For each object in ['French chef in a Native', ' American headdress is wrestling', ' African tribal chiefs', ' Russian Orthodox church'], if that object and a certain culture in the [France, America, Russia] conflict with each other, please output it. The format is "Something should not be in Some Culture". For example, "Katana should not be in America", "Chinese calligrapher should not be in Africa". If there isn't contradict, output nothing.
**Ground-truth Answer**: "French chef in a Native should not be in United States.", · · · .
**Output**:Something should not be in Some Culture.

Figure 16: **An example of "Stubbornness" is presented.** The MLLM fails to generate the required substitutions, leaving both the placeholder terms 'Something' and 'Some Culture' unmodified in its output.

Sure, here are the translations of the sentences you provided:
1. kuru na!: This is an interjection used to express excitement or surprise, similar to "Wow!" or "Whoa!" in English.
2. kuru na!: This is another way of expressing excitement or surprise, similar to "Wow!" or "Whoa!"

Figure 17: **An example of poor instruction following capability of MiniGPT**. Our instruction clearly required MiniGPT to directly output the target translations without additional content. Nevertheless, the model responded with redundant descriptive text (e.g., explaining the sentence type and emotional connotation) and duplicated translations for the identical Japanese input. Both issues indicate its failure to adhere to the constraint of instruction.

From the results, it is evident that OCR significantly degrades model performance. Since we did not utilize OCR in our experiments, the observed lower model performance is not attributable to OCR-related factors.

Table 8: **Main Result of effect of OCR on MT tasks**. The highest performance is marked in **bold**.

| Methods | JA-DE | | JA-RU | | JA-ES | | JA-TH | |
|---|---|---|---|---|---|---|---|---|
| | OCR | No OCR | OCR | No OCR | OCR | No OCR | OCR | No OCR |
| Qwen2.5-VL | 0.92 | **12.0** | 0.97 | **8.74** | 1.32 | **14.5** | 1.22 | **9.72** |
| **LLaVA1.5-7B** | 0.64 | **2.74** | 0.24 | **1.39** | 0.62 | **4.23** | 0.32 | **1.13** |
| **Llama3.2** | 0.33 | **4.27** | 0.22 | **1.70** | 0.45 | **5.73** | 2.73 | **0.99** |