# OpenReview forum: "Culture In a Frame: C$^3$B as a Comic-Based Benchmark for Multimodal Culturally Awareness"
_ICLR.cc/2026/Conference — ICLR 2026 Poster_

### Official Review · Reviewer_6ZTX · 2025-10-28

**Soundness:** 4
**Presentation:** 2
**Contribution:** 3
**Rating:** 8
**Confidence:** 4

**Summary:**

This paper introduces C³B (Comics Cross-Cultural Benchmark), a novel benchmark aimed at evaluating Multimodal Large Language Models (MLLMs) for their cultural awareness capabilities across multiple languages and cultural contexts. Unlike prior datasets that rely on real-world images—each typically representing a single cultural context—C³B uses comics as the core medium. The authors argue that comics can depict multiple, fictional, and mixed cultural elements within a single frame, thus presenting richer and more complex cross-cultural scenarios.

C³B evaluates:
1. Culture-aware Object Extraction (Extraction@Culture) – basic recognition of cultural cues and artifacts;
2. Cultural-conflict Object Detection (Conflict@Culture) – detecting contradictions between cultural elements;
3. Culturally-aligned Content Generation (Generation@Culture) – multilingual cultural translation tasks (Japanese → EN/RU/DE/TH/ES).

The benchmark was used to evaluate 11 open-source MLLMs (e.g., Qwen2.5-VL, LLaVA-NeXT, InternLM-XC2.5, Llama3.2). Results indicate that Qwen2.5-VL performs best overall, particularly in multilingual translation, but all models fall far behind human performance—especially in understanding cultural conflicts and less-known cultures. Human evaluation shows a consistent 40–60% gap over MLLMs, confirming C³B’s difficulty and diagnostic potential.

**Strengths:**

### Innovative use of comics as a medium
The idea of using comics for cross-cultural benchmarking is both creative and effective. Comics enable a richer mixture of visual and textual cues that naturally embed multiple cultures, allowing for deeper testing of model reasoning and perception.

### Progressive multitask structure
The three-level design (extraction → conflict → generation) provides a structured evaluation pipeline that mimics human-like cultural reasoning progression—from perception to comprehension to production.

### Multilingual and multicultural scope
C³B covers 77 cultures and 5 languages, far broader than prior cultural benchmarks like CVQA or GIMMICK. The inclusion of low-resource languages such as Thai enhances diversity and fairness.

### Detailed methodological design
The construction pipeline—using multi-agent annotation with both human verification and LLM assistance (DeepSeek-V3)—is well-documented and reproducible. Figures 2–3 (pp. 4–5) clearly illustrate the dataset creation process.

### Comprehensive evaluation and analysis
The experiments include cross-task correlation analysis (Table 5), cultural diversity metrics (Table 6), and per-culture accuracy (Figure 5). These analyses go beyond standard benchmarking, offering valuable insights into model cultural biases and limitations.

**Weaknesses:**

### Limited evaluation diversity in the generation task
The Generation@Culture task focuses solely on machine translation. This limits the benchmark’s exploration of more nuanced cultural generation (e.g., culturally appropriate narration, idiomatic expression, or story understanding).

### Use of AI-assisted annotation may introduce bias
Although manually verified, the use of DeepSeek-V3 for conflict detection and translation generation could embed model-specific biases, especially in culturally sensitive interpretations.

**Questions:**

Since DeepSeek-V3 was used in annotation, did you compare its conflict-detection output to human-only annotations to quantify model bias?
Have you performed any human evaluation of cultural fidelity (i.e., are the depicted elements realistic to their culture)?

overall, it's a good paper.

---

> ### Author Response · Authors · 2025-11-20
> **Response to Reviewer 6ZTX**
>
> **Q1**: The Generation@Culture task focuses solely on machine translation. This limits the benchmark’s exploration of more nuanced cultural generation (e.g., culturally appropriate narration, idiomatic expression, or story understanding).
>
> **A1**: We believe that evaluating the Generation@Culture task via machine translation (MT) is justified for the following reasons:
>
> First, the alternative task types—including culturally appropriate narration, idiomatic expression generation, and story understanding—have significant limitations. While story understanding has been adopted in relevant literature [1-3] as a measure of MLLMs’ comprehension capabilities, culturally appropriate narration and idiomatic expression generation lack precise reference texts, making objective and accurate evaluation infeasible.
>
> In contrast, the MT task effectively addresses these drawbacks. For one, MT has long been a well-established and extensively studied generation task in NLP [4-6], providing a mature evaluation paradigm. For another, with manually annotated reference translations, we can leverage multiple quantitative metrics (e.g., BLEU, COMET) to conduct rigorous and interpretable evaluations.
>
> Furthermore, cultural-related tasks should not be confined to the relationship between multimodality and English; they must also account for the impact of multilingualism on cross-cultural understanding. However, multilingualism has not been widely explored in tasks like image captioning or cultural storytelling. Considering this critical dimension, the MT task emerges as a natural and robust choice for our benchmark.
>
> In summary, adopting MT as the generation task in Generation@Culture is reasonable.
>
> [1] SEED-Story: Multimodal Long Story Generation with Large Language Model, ICCV 2025
>
> [2] StoryLLaVA: Enhancing Visual Storytelling with Multi-Modal Large Language Models. ACL 2025.
>
> [3] TimeChat: A Time-sensitive Multimodal Large Language Model for Long Video Understanding. CVPR 2020.
>
> [4] Survey of Low-Resource Machine Translation. Computational Linguistics 48.3 (2022).
>
> [5] Bridging Sparse Domain Semantics via an Asymmetric Siamese Framework with Virtual Anchor Guidance for Domain-Specific Multimodal Translation. Artificial Intelligence (2025): 104443.
>
> [6] Enhancing Entertainment Translation for Indian Languages Using Adaptive Context, Style and LLMs. AAAI 2025.
>
> **Q2**: Although manually verified, the use of DeepSeek-V3 for conflict detection and translation generation could embed model-specific biases, especially in culturally sensitive interpretations.
>
> **A2**: We have incorporated manual annotation throughout multiple key steps, including question design, annotation of cultural contexts and cultural objects in images, and labeling of ground-truth translations. Moreover, as detailed in Lines 991–994 of Appendix E, expert review was integrated into steps of the text-to-image generation process. This rigorous process is intended to minimize bias and avoid sensitive interpretations to the greatest extent possible.
>
> **Q3:** Since DeepSeek-V3 was used in annotation, did you compare its conflict-detection output to human-only annotations to quantify model bias?
>
> **A3**: As noted in Lines 264–269 of the manuscript, during the annotation process for the Conflict@Culture task, we manually reviewed and validated the outputs generated by DeepSeek-V3. This human-in-the-loop quality control step ensured the accuracy and reliability of the final annotations.
>
> **Q4**: Have you performed any human evaluation of cultural fidelity (i.e., are the depicted elements realistic to their culture)?
>
> **A4**: We incorporated manual annotation throughout the entire workflow—from text creation to image generation. The primary objective of this annotation process is to ensure that both the textual content and generated images convey authentic cultural information. Any text or images deemed insufficiently authentic were excluded from the final dataset.
>
> **Q5**: overall, it's a good paper.
>
> **A5**: Thank you for your recognition! Comics are an incredibly engaging modality that offers rich potential for exploring cultural diversity. We will continue to delve deeper into cultural-related tasks and pursue more explorations in generative tasks. Your suggestions are greatly appreciated!

---

> > ### Comment · Reviewer_6ZTX · 2025-11-26
> > **Response**
> >
> > The reponse solved my problem partly, but still, I believe there's much improvement room need to be done further regarding the MT problem, I will keep my positive score for the current paper.

---

> > > ### Author Response · Authors · 2025-11-27
> > > **Thanks for your support!**
> > >
> > > We sincerely appreciate your valuable feedback and support, and we are pleased to learn that the majority of your concerns have been effectively addressed. Regarding the MT problem, we believe the results show that a well-designed cultural generation task is inherently challenging. Inspired by your insights in Q1, we agree that culturally appropriate story generation is a task worth further discussion. It covers all three aspects you mentioned: culturally appropriate narration, idiomatic expression, and story understanding. Thank you again for your valuable comment.

---

### Official Review · Reviewer_MQS5 · 2025-10-31

**Soundness:** 3
**Presentation:** 3
**Contribution:** 4
**Rating:** 6
**Confidence:** 4

**Summary:**

The paper introduces a comics cross-cultural benchmark ($\mathrm{C}^3$B) designed to measure MLLMs' cultural awareness beyond Western-centric settings. Motivated by limits in prior datasets, including single culture and real image focus, one-question tasks, and weak multilingual coverage, $\mathrm{C}^3$B uses fictional comic scenes to pack multiple cultures into a single frame and raise task difficulty. The benchmark contains 2,220 images and 18,789 QA pairs spanning 77 cultures, aiming to evaluate models in multicultural, multitask, and multilingual contexts.

$\mathrm{C}^3$B comprises three progressively harder tasks forming a logical chain: culture-aware object extraction, cultural-conflict detection, and culturally-aligned content generation. Evaluating 11 open-source MLLMs on $\mathrm{C}^3$B, the authors find a clear gap between models and humans, especially on cultural-conflict reasoning.

**Strengths:**

- Solid benchmark scale and diversity (2,220 images, 18,789 QAs, and 77 cultures).
- Annotation pipeline combines automated detection with manual verification.
- Multilingual coverage across five languages (Japanese, Russian, Thai, English, and Spanish) reduces monolingual bias.

**Weaknesses:**

- Frame-level evaluation may underutilize narrative or contextual reasoning across pages.
- CACC uses fixed weights (0.3 / 0.3 / 0.4) without dynamic justification.
- Comics differs from real-world photos, making transfer to natural imagery uncertain.

**Questions:**

- How well do the skills measured on $\mathrm{C}^3$B transfer to real photos? Have you conducted any cross-benchmark evaluations?
- Given that Doubao generated the comic pages and DeepSeek-V3 handled conflict labels and translations, what measures did you take to prevent bias or information leakage from those models?
- What was the justification for selecting the 0.3/0.3/0.4 weights for Q1/Q2/Q4?

---

> ### Author Response · Authors · 2025-11-20
> **Response to Reviewer MQS5 (Part 1)**
>
> **Q1**: Frame-level evaluation may underutilize narrative or contextual reasoning across pages.
>
> **A1**: First, among the three tasks we designed, the first two primarily focus on evaluating MLLMs’ visual recognition capabilities. To this end, we adopted a frame-level design—this is because we need to ensure MLLMs accurately recognize elements within each individual frame.
>
> Second, in Generation@Culture, we utilized real comic images, which are typically multi-frame. This setup allows us to assess MLLMs’ capabilities in an across-page context.
>
> Finally, your comment has inspired us: in future work, the input information for cultural benchmarks should be more comprehensive. For instance, entire comic books could be used as input to enable the evaluation of MLLMs’ abilities in chapter-level cultural understanding of comics, among other relevant competencies.
>
> **Q2**: CACC uses fixed weights (0.3 / 0.3 / 0.4) without dynamic justification.
>
> **A2**: Yes, this weight distribution was intentionally designed. We believe Q1, Q2, and Q4 are roughly comparable in importance, but the performance of Q4 should be the dominant factor in the evaluation.
>
> To further address your concern, we conducted supplementary experiments using alternative static weight configurations. The results are presented below: (Experiment Setting: Q4 is dominant, with its weight greater than that of Q1 and Q2 (all weights are presented as one decimal place).)
>
> | **Weight(Q1/Q2/Q4)** | **CACC** |
> | --- | --- |
> | 0.1 0.1 0.8 | 7.47 |
> | 0.1 0.2 0.7 | 10.28 |
> | 0.1 0.3 0.6 | 13.08 |
> | 0.1 0.4 0.5 | 15.89 |
> | 0.2 0.1 0.7 | 10.80 |
> | 0.2 0.2 0.6 | 13.60 |
> | 0.2 0.3 0.5 | 16.41 |
> | 0.2 0.4 0.4 | 19.22 |
> | 0.3 0.1 0.6 | 14.13 |
> | 0.3 0.2 0.5 | 16.93 |
> | **0.3 0.3 0.4** | **19.74** |
> | 0.4 0.1 0.5 | 17.46 |
>
> The experimental results indicate that while CACC scores vary numerically across different weight combinations, the overall trend remains consistent. The default weights adopted in this study (0.3 / 0.3 / 0.4) achieved the highest comprehensive score among all configurations and most clearly distinguished the difficulty gap between Q4 and Q1/Q2. Consequently, this design not only aligns with the task dependency structure (where Q4 builds on Q1/Q2) but also empirically delivers the most stable and discriminative evaluation outcomes. Overall, the current hyperparameter selection is reasonable and robust.
>
> **Q3**: Comics differs from real-world photos, making transfer to natural imagery uncertain.
>
> **A3**: In fact, comics have long been a real-world medium—even before the widespread adoption of photography, illustrations and comics served as a primary means to document reality. As humans can recognize, understand, and generate cultural content through comics, we aim to evaluate whether current MLLMs can achieve the same capability. This motivated the construction of our benchmark.
>
> To address your question more comprehensively, we conducted a supplementary experiment: we fine-tuned the Qwen-2.5-VL model using C3B (focusing on Q1), and then evaluated its performance on CVQA[1]. The results are presented below:
>
> | **Method** | **ACC** |
> | --- | --- |
> | Qwen-2.5-VL w/o C3B | 19.6 |
> | Qwen-2.5-VL w/ C3B | **27.1** |
>
> The experimental results demonstrate that using C3B as training data **significantly** improves the model’s performance on the real-world photo benchmark—providing strong evidence for the excellent generalization ability of our dataset.
>
> [1] CVQA: culturally-diverse multilingual visual question answering benchmark. NeurIPS. 2024.
>
> **Q4**: How well do the skills measured on B transfer to real photos? Have you conducted any cross-benchmark evaluations?
>
> **A4**: As demonstrated by the Experiment Q3, fine-tuning large language models (MLLMs) with the C3B dataset can also enhance the model’s ability on CVQA.
>
> **Q5**: Given that Doubao generated the comic pages and DeepSeek-V3 handled conflict labels and translations, what measures did you take to prevent bias or information leakage from those models?
>
> **A5**: First, we ensured through manual annotation that each image contains authentic cultural elements corresponding to the indicated cultural group. Additionally, both the design of questions and the formulation of answers involved human curation—this rigorous process allows us to mitigate the risks of bias and data leakage effectively.
>
> Detailed information about our annotation workflow has been supplemented in the Appendix E; please refer to it for further clarification.

---

> ### Author Response · Authors · 2025-11-20
> **Response to Reviewer MQS5 (Part 2)**
>
> **Q6**: What was the justification for selecting the 0.3/0.3/0.4 weights for Q1/Q2/Q4?
>
> **A6**: Since Q4 requires responses based on Q1 and Q2, we needed to account for the influence of Q1 and Q2 on Q4 when designing the evaluation framework. Consequently, the weights were calibrated to ensure adequate consideration of Q1 and Q2 while assigning a relatively higher proportion to Q4’s accuracy—this was the rationale behind our weight design.
>
> As further validated by the results of Experiment Q2, the rationality of our parameter settings is well-supported.

---

> ### Author Response · Authors · 2025-11-27
> **Gentle Reminder for Reviewer MQS5**
>
> Dear Reviewer MQS5,
>
> We would like to express our sincere gratitude once again for your review of our submission.
>
> In our prior response, we addressed your concerns point-by-point, including those related to fixed weights in CACC (Q2, Q6), transferability to real photos (Q3, Q4), frame-level evaluation (Q1), and bias mitigation details (Q5).
>
> To further alleviate your concerns regarding transferability to real photos, we have fine-tuned additional models on the C3B benchmark and evaluated their performance on the CVQA dataset. The comprehensive results are presented as follows:
>
> | **Method** | **ACC** |
> | --- | --- |
> | Qwen-2.5-VL w/o C3B | 19.6 |
> | Qwen-2.5-VL w/ C3B | **27.1** |
> | Llama3.2 w/o C3B | 24.1 |
> | Llama3.2 w/C3B | **28.9** |
> | InternVL w/o C3B | 26.0 |
> | InternVL w/ C3B | **26.6** |
>
> The results demonstrate that fine-tuning with the C3B benchmark effectively enhances model performance on tasks involving real images, providing direct evidence for the transferability.
>
> We stand ready to engage in further discussion. Your insights are highly valuable to refining our work. We eagerly anticipate your feedback!
>
> Best Regards,
>
> All Authors

---

> ### Comment · Reviewer_MQS5 · 2025-11-27
>
> Thank you for the detailed response and for addressing my comments. I appreciate the additional experiments, and I have raised my score accordingly.

---

> > ### Author Response · Authors · 2025-11-27
> > **Thanks for your support!**
> >
> > Thank you very much for your kind feedback and support! We are glad to hear that your concerns have been addressed.

---

### Official Review · Reviewer_oEmH · 2025-10-31

**Soundness:** 2
**Presentation:** 1
**Contribution:** 2
**Rating:** 2
**Confidence:** 4

**Summary:**

The paper introduces C3B (Comics Cross-Cultural Benchmark), a comic-centric evaluation suite for cultural awareness in MLLMs. C3B contains 2,220 images and 18,789 QA pairs, spanning three escalating tasks: (i) culture and object detection, (ii) cultural-conflict object detection and description, and (iii) machine translation. The dataset mixes generated comics via Doubao APIs with automatic and manual annotations of culturally representative objects and conflicts (DeepSeek-V3-assisted) and Manga109 pages for translation with a translator–reviewer multi-agent pipeline for references. The authors benchmark 11 open-source MLLMs, report low performance, especially on conflict description (Q4), and show a consistent gap vs. human accuracy across difficulty tiers.

**Strengths:**

- Framing culture evaluation around comics is novel: unlike many real-world images that encode a single culture, comics can “condense numerous cultures into a single frame”, which plausibly raises task difficulty, but also introduces the problem of artificially generated data that do not depict real-world needs.
- The dataset combines multitask + multicultural + multilingual integration and progressive difficulty. Something that is not clarified very well in the paper is that in C3B, there are two different dimensions of progressive difficulty. One is in the three-level structure (recognition → conflict reasoning → multilingual generation), which provides a progressive, multitask view missing in several prior cultural benchmarks that are monolithic or single-task. The second one is in the number of cultural objects/conflicts identified in one image.

**Weaknesses:**

- Problems with tasks, prompt set-up, and phrasing. The prompts, as depicted in the Figures, contain grammatical mistakes and appear to be too complicated. See questions/comments for further details. I would recommend actually phrasing the first task as recognition (what cultures/objects appear as multiple choice (q1) and open-ended (q2) questions). Then, I would rename cultural conflict understanding to cultural conflict reasoning, emphasizing that the goal is not merely to detect a contradiction (q3), but to reason about its cause and cultural context (q4). For the Generation@Culture task, I would reconsider using machine translation as the generative component. While translation can reflect cultural nuances, it does not fully test a model’s ability to generate culturally informed or contextually appropriate content. Instead, I suggest reframing this task toward culture-aware image captioning or cultural storytelling, where models must produce natural-language descriptions or short narratives that reflect the depicted cultures’ symbols, values, or interactions. This adjustment would make the task hierarchy more coherent (recognition → reasoning → generation) and align better with standard multimodal evaluation practices.
- Problems with dataset bias and validation. A significant share of C3B relies on generated comics and LLM-mediated conflict labels. This can introduce artifact-driven shortcuts or stereotype leakage from generators/LLMs—risks widely discussed for culture benchmarks and synthetic imagery. The paper partially mitigates with manual verification, but a deeper bias and authenticity audit (e.g., culture-expert review, stereotype checklists) would strengthen claims about real-world cultural awareness.
- Problems with annotation quality and manual validation. The paper does not clearly describe how annotators (models, agents, or authors) were instructed to identify cultures, objects, or cultural conflicts, nor how manual validation was performed or measured (e.g., number of validators, agreement scores, or correction rate). Please provide explicit annotation details and quality-control metrics to ensure transparency and reproducibility.
- Problems with ground truth for conflict reasoning. Q4 depends on Q1/Q2 labels plus model answers; the paper reports a composite CACC with fixed weights. While motivated, this mixes upstream recognition errors with conflict reasoning, and the ablation shows only marginal gains when injecting Q1/Q2 answers directly into the Q4 prompt. Stronger gold annotations for conflicts (independent of model responses) and sensitivity checks on CACC weights would clarify what Q4 truly measures.

**Questions:**

- How were stereotypes and sensitive portrayals handled during generation and annotation? Any expert review (region-specific annotators) or exclusion lists? A brief ethics appendix would help, given synthetic cultural content concerns.
- For Q4, do you have independent gold conflict triples (culture, object, contradiction) for a subset, rather than relying on model-conditioned answers? If not, could you release human-vetted conflict graphs to decouple recognition errors from reasoning?
- The human evaluation study needs further elaboration. You bucket difficulty by the number of cultures per image, then sample 100 items per tier. Could you add inter-rater agreement and per-tier time-to-answer? Also, show model vs. human delta broken down by specific culture sets (e.g., lesser-known vs. globally familiar).
- Background Culture Identification and Culture-aware Object Detection: How are gold labels established? Please describe annotator guidelines, the source of culture/object lists, and any expert review. Include inter-annotator agreement (e.g., Cohen’s κ) and examples of borderline cases. Explicitly discuss bias mitigation (e.g., stereotype checks, exclusion lists, region-specific reviewers).
- How do you define cultural conflict? Provide an explanation and decision rules with concrete examples from the annotation/internal validation process and guidelines. Clarify whether conflicts are culture–culture, culture–object, or object–object, and how multi-culture scenes are handled.
- Clarify whether the Generation@Culture task uses OCR, classic machine translation, or general-purpose generative LLMs for translation. Is the input the manga image or the transcription of the text presented in the image?
- Low BLEU: disentangle error sources. Are low scores primarily due to translation quality or OCR extraction errors? If OCR is used on Manga109, I would recommend first reporting OCR accuracy or manual correction rates. State whether you have ground-truth transcriptions for source text from Manga109; if yes, use them to isolate translation quality from text extraction noise.

**Grammar mistakes/Wording fixes**
- Line 13: “Cultural awareness capabilities has emerged” → “Cultural awareness capabilities have emerged.”
- “current benchmarks lack progressed difficulty?” → It is unclear what is meant by progressed difficulty in the abstract, maybe progressive? I would rephrase: “Existing benchmarks do not offer a clearly staged progression of difficulty.” or something that shows that you mean progressively challenging tasks.
- Lines 125: “integrates comics and comicss” →  Do you mean and manga? It is not clear.
Lines 249-251: This is not good English and is confusing to the reader. "Annotation for task..." -> "The annotation for the tasks of" AND "For task Extraction@Culture" -> "For the task of Extraction@Culture"
- Lines 260–269: I would recommend avoiding the future tense in the methods section. For example, replace “We will manually inspect…” with “We manually inspect (or inspected)…” (past or present).
- Lines 420–424: Rewrite for grammar and specificity. I assume you mean: “To evaluate C3B’s cultural diversity, we compute three measures—Culture Density Per Image (CDPI), Cultural Breadth Intensity (CBI), and Coverage-Adjusted Density (CAD)—and compare them against cultural QA datasets.”
- Figure 1: “Is there a culture contradicts in this image?” → “Does this image contain a cultural contradiction?”, “If there is contradict in this image…” → “If a contradiction is present, answer the following questions.”, “For each object in the answer to your question 2 -/ contradiction!!” → Please rephrase the instruction clearly; e.g., “For each object identified in Q2, explain the cultural contradiction it creates.”, Labeling error: Use “C3B” instead of “C3UB.”
- Figure 2: Same labeling error—“C3B,” not “C3UB.”
- Table 3: Again, “C3B,” not “C3UB.”
- Figure 5 caption: “Scores of QA Pairs in different culture” is ambiguous. Specify the metric: “Error rates of QA pairs by culture”. Also define the denominator (per pair? per culture? macro/micro?).

**Comments**
- Given that culture-related benchmarks are still relatively scarce, I recommend expanding the Related Work 2.2 section to include recent datasets that explicitly address cultural diversity in multimodal evaluation. For instance, the VizWiz cultural extension ([Karamolegkou et al., 2024](https://aclanthology.org/2024.hucllm-1.5.pdf)) demonstrates that existing real-world benchmarks, such as VizWiz, encode cultural concepts that were not taken into account in the original crowdsourced annotations. Thus, it introduces a culture-centric evaluation subset in which some images are annotated with more than one culture (e.g., a US brand package of matcha tea, a French wine packaged by Tesco in the UK, containers of Asian food from an Australian meal delivery company, etc.). This work is relevant to your focus on cross-cultural visual understanding and includes images taken in real-world settings by people who are blind that reference more than one culture. Another relevant work that seems to be missing is GlobalRG ([Bhatia et al., 2024](https://aclanthology.org/2024.emnlp-main.385.pdf)), which introduces a multicultural benchmark and two novel tasks—Retrieval across Universals and Cultural Visual Grounding—designed to assess vision-language models' performance across culturally diverse and culture-specific concepts. Including this paper would strengthen the discussion on cultural inclusivity and dataset diversity in VLM evaluation. In addition, ALM-bench ([Vayani et al., 2025](https://arxiv.org/pdf/2411.16508)) represents the largest-scale effort to date to evaluate multimodal models across 100 languages and 13 cultural dimensions, explicitly measuring cultural and linguistic inclusivity. Its scope and methodological framing make it a strong point of comparison for your benchmark. You may also consider referencing [Zhong et al., 2025](https://arxiv.org/html/2510.05931v2), which introduces a complementary perspective on multimodal cultural reasoning and could help position C3B more clearly within the rapidly growing ecosystem of cross-cultural LMM evaluation.
- Lines 270–280: I would recommend adding a quantitative summary of that inspection (e.g., number reviewed, acceptance rate, inter-rater agreement).
- For instruction-following issues (e.g., “stubbornness” in LLaVA-1.5-7B), this is a prompt compliance problem rather than cultural understanding. Consider simple schema validation (e.g., JSON schema with pydantic) to enforce output format and reduce evaluation noise; note it as an implementation detail, not a scientific claim.
- For cultural density measures (CDPI/CBI/CAD), add brief intuition + formulae in main text and validate that higher density correlates with lower model accuracy on held-out items.
- If you claim “progressive difficulty,” define how difficulty is constructed and validated (e.g., number of cultures per image, ambiguity, required cross-culture reasoning) and show that model performance monotonically decreases with the proposed difficulty index (or explain deviations). You should also clarify that by progressive difficulty, you also include the dimension of how many cultural objects are contained in one image (as it is only briefly mentioned in the human-model analysis).
- To compensate for the limitations of synthetic data and artificial tasks, you could draw inspiration from [Evaluating Multimodal Language Models as Visual Assistants for Visually Impaired Users](https://aclanthology.org/2025.acl-long.1260/) (Karamolegkou et al., ACL 2025). That work grounds multimodal evaluation in real-world needs, such as question answering and image captioning for accessibility. You should acknowledge it in related works, since MLMMs are now being used in the real world. Incorporating similarly practical, human-centered tasks would make C3B more ecologically valid and demonstrate the benchmark’s relevance beyond synthetic cultural scenarios.

---

> ### Author Response · Authors · 2025-11-20
> **Response to Reviewer oEmH (Part 1)**
>
> **Q1-1**: Problems with tasks, prompt set-up, and phrasing. The prompts, as depicted in the Figures, contain grammatical mistakes and appear to be too complicated.
>
> **A1-1**: Actually, the prompt design in our paper is reasonable, and we address your concerns as follows: Regarding the perception that the prompts are "too complicated," existing research on prompt engineering [1-3] has shown that detailed prompts supplemented with examples can help avoid misunderstandings by LLMs and improve the accuracy of their outputs.
>
> As for potential grammatical issues: we used the same prompt consistently across all tasks in our experiments, which ensures the fairness of the experimental setup. To further alleviate your concern, our team has reviewed the prompts and confirmed that there are no issues affecting the core semantics of the instructions. Nevertheless, we have polished the prompts for clarity, and the revised versions are presented below:
>
> ```jsx
> Q1: Which culture(s) does the background in the comics page belong to? Select all possible answers from the provided options.
>
> Q2: Which object in the picture reflects a specific cultural attribute? Select the most culturally representative and appropriate answer from the provided options.
>
> Q3: Please answer the following question based on the provided comics image: Does the image contain any cross-cultural contradictions? Provide a binary answer ("Yes" or "No").
>
> Q4: Check the \(A_2\) object (from Question 2) against \(A_1\) cultures (from Question 1). Output conflicts strictly as: "[Object] should not be in [Culture]" (e.g., "Katana should not be in American culture"). Output nothing if no conflict or Question 3’s answer is "No".
> ```
>
> Using the revised prompts, we conducted experiments on Qwen2.5-VL for Questions 1–4 (Q1–Q4). The results are presented as follows:
>
> |  | **Q1** | **Q2** | **Q3** | **Q4ACC** | **Q4CACC** |
> | --- | --- | --- | --- | --- | --- |
> | **Qwen2.5-vl (Original Prompts)** | 53.7 | 55.9 | 63.1 | 3.20 | 34.2 |
> | **Qwen2.5-vl (Modified Prompts)** | 53.8 | 53.8 | 50.4 | 3.17 | 33.5 |
>
> The results indicate that after revising the relevant content of the prompts, there is no significant change in the experimental outcomes, which remain consistent with our current conclusions.
>
> [1] Exploring the relationship between in-context learning and instruction tuning. EMNLP 2024.
>
> [2] Coverage-based Example Selection for In-Context Learning. EMNLP 2023.
>
> [3] Why can GPT learn in-context? language models secretly perform gradient descent as meta-optimizers. ACL 2023.
>
> **Q1-2**: See questions/comments for further details. I would recommend actually phrasing the first task as recognition (what cultures/objects appear as multiple choice (q1) and open-ended (q2) questions).
>
> **A1-2**: As specified in the current manuscript, the first task is designed as a recognition task. To evaluate both the ability to understand the cultures depicted in the images and the ability to extract cultural elements, we employed two question formats: multiple-choice questions (Q1) and open-ended questions (Q2).
>
> **Q1-3**: Then, I would rename cultural conflict understanding to cultural conflict reasoning, emphasizing that the goal is not merely to detect a contradiction (q3), but to reason about its cause and cultural context (q4).
>
> **A1-3**:  We clarify that Questions 3 and 4 (Q3 and Q4) are designed to evaluate MLLMs’ inherent ability to understand cultural conflicts. In contrast, the reasoning component—building on the premise that MLLMs already possess such understanding—aims to further assess their capacity to generalize to diverse cultural conflict scenarios through reasoning [1-3]. Thus, the current positioning of Q3 and Q4 is justified.
>
> On this point, our understanding aligns with that of Reviewer 6ZTX: Q3 and Q4 embody a progression from perception to comprehension to production.
>
> [1] Commonsense Reasoning in Arab Culture. CORR 2025.
>
> [2] IndoCulture: Exploring Geographically Influenced Cultural Commonsense Reasoning Across Eleven Indonesian Provinces. TACL 2024.
>
> [3] SeaEval for Multilingual Foundation Models: From Cross-Lingual Alignment to Cultural Reasoning. NAACL 2024.

---

> ### Author Response · Authors · 2025-11-20
> **Response to Reviewer oEmH (Part 2)**
>
> **Q1-4**: For the Generation@Culture task, I would reconsider using machine translation as the generative component. While translation can reflect cultural nuances, it does not fully test a model’s ability to generate culturally informed or contextually appropriate content. Instead, I suggest reframing this task toward culture-aware image captioning or cultural storytelling, where models must produce natural-language descriptions or short narratives that reflect the depicted cultures’ symbols, values, or interactions. This adjustment would make the task hierarchy more coherent (recognition → reasoning → generation) and align better with standard multimodal evaluation practices.
>
> **A1-4**: We believe that evaluating our Generation@Culture task via machine translation (MT) is justified, and we elaborate on the rationale as follows:
>
> First, two alternative tasks—image captioning and cultural storytelling—have significant limitations. Image captioning has been used in some related works [1-5] to reflect MLLMs’ comprehension abilities, but most of the captions generated by MLLMs are relatively short.  Cultural storytelling lacks precise reference texts, making accurate evaluation infeasible. In contrast, the MT task effectively addresses both limitations: on one hand, MT has long been an extensively studied generative task in NLP [6-8], with well-established evaluation paradigms; on the other hand, the availability of human-annotated reference translations enables quantitative assessment using multiple metrics such as BLEU and COMET.
>
> Second, cultural tasks should not be limited to the relationship between multimodality and English—they must also consider the impact of multilingualism on cross-cultural understanding. However, multilingualism has not been widely explored in image captioning or cultural storytelling. Given this critical dimension, the MT task emerges as a natural and appropriate choice.
>
> In summary, using MT as the generative task for our Generation@Culture task is both reasonable and well-justified.
>
> Finally, Reviewer 6ZTX, we share your view that cultural generation warrants more in-depth discussion in future research.
>
> [1] Image Captioning Evaluation in the Age of Multimodal LLMs: Challenges and Future Perspectives." IJCAI 2025.
>
> [2] Personalizing Multimodal Large Language Models for Image Captioning: An Experimental Analysis. ECCV 2024.
>
> [3] Multimodal attention with image text spatial relationship for ocr-based image captioning. ACM MM 2020.
>
> [4] Sieve: Multimodal dataset pruning using image captioning models. CVPR 2024.
>
> [5] Polos: Multimodal metric learning from human feedback for image captioning. CVPR 2024.
>
> [6] Survey of Low-Resource Machine Translation. Computational Linguistics 48.3 (2022).
>
> [7] Bridging Sparse Domain Semantics via an Asymmetric Siamese Framework with Virtual Anchor Guidance for Domain-Specific Multimodal Translation. Artificial Intelligence (2025): 104443.
>
> [8] Enhancing Entertainment Translation for Indian Languages Using Adaptive Context, Style and LLMs. AAAI 2025.
>
> **Q2-1**: Problems with dataset bias and validation. A significant share of C3B relies on generated comics and LLM-mediated conflict labels. This can introduce artifact-driven shortcuts or stereotype leakage from generators/LLMs—risks widely discussed for culture benchmarks and synthetic imagery.
>
> **A2-1**:  In fact, our paper provides sufficient discussion on the generated comics and LLM-mediated conflict labels. During the annotation process, manual evaluation was conducted at each step to ensure the relevance and accuracy between the images and the corresponding questions.
>
> Moreover, existing works that utilize AI-generated images as benchmarks have been extensively explored [1-2], which validates the rationality of the settings adopted for content generation in our study. Furthermore, our dataset is not solely composed of generated content, because we also incorporated real-world comic data. To mitigate potential biases introduced by comic styles, we generated comics in two distinct styles: Japanese (accounting for 76.44%) and American  (23.56%), with real comic data making up one-third of the entire dataset. Overall, our dataset composition is well-justified and avoids significant biases. Additional statistical details regarding the dataset images are provided in Table 2.
>
> Regarding the concern of stereotype leakage: as presented in the manual experiment in Section 5.6 of the paper, human performance significantly outperforms that of MLLMs. If stereotype leakage had occurred, the MLLMs’ performance would not have been as low as observed.
>
> [1]: Journeydb: A benchmark for generative image understanding. NeurIPS 2023
>
> [2] LLaVA-Video: Video Instruction Tuning With Synthetic Data, TMLR, cited 351.

---

> ### Author Response · Authors · 2025-11-20
> **Response to Reviewer oEmH (Part 3)**
>
> **Q2-2:** The paper partially mitigates with manual verification, but a deeper bias and authenticity audit (e.g., culture-expert review, stereotype checklists) would strengthen claims about real-world cultural awareness.
>
> **A2-2**: Our overall dataset construction is grounded in contributions from native culture-expert review and their culture-specific knowledge. Consequently, cultural knowledge errors and biases have been avoided even during the image and question validation phase. Detailed descriptions of this process are provided in Lines 1004–1008 of the manuscript.
>
> **Q3**: Problems with annotation quality and manual validation. The paper does not clearly describe how annotators (models, agents, or authors) were instructed to identify cultures, objects, or cultural conflicts, nor how manual validation was performed or measured (e.g., number of validators, agreement scores, or correction rate). Please provide explicit annotation details and quality-control metrics to ensure transparency and reproducibility.
>
> **A3**: We have detailed how we guided human annotators to identify cultures and cultural objects, as well as how we excluded images containing aggressive content, in Lines 968–971 and Lines 991–994 of the manuscript. Additionally, we noted in Lines 960–962 that our annotation team comprises **five** human annotators.
>
> To enhance transparency, we have supplemented the annotation details: as specified in Lines 996–997, our inter-annotator agreement score is **76.8%**, and Lines 1001–1002 indicate a correction rate of **73.8%**. Following these steps, a final manual review was conducted (see Lines 1004–1007) to further ensure the accuracy of our annotations.
>
> To address your concern more thoroughly, we have refined the description of the annotators’ background and expertise, which has been updated in the Appendix E.
>
> **Q4-1**: Problems with ground truth for conflict reasoning. Q4 depends on Q1/Q2 labels plus model answers; the paper reports a composite CACC with fixed weights.
>
> **A4-1:** Actually, the weight setting for CACC in our experiments is well-justified. The current weights of CACC are designed to better distinguish the relative impact of Q1/Q2 and Q4. To address your concern, we conducted ablation studies on the selection of CACC weights. The experimental results are presented as follows: (Experiment Setting: Q4 is dominant, with its weight greater than that of Q1 and Q2 (all weights are presented as one decimal place.)
>
> | **Weight(Q1/Q2/Q4)** | **CACC** |
> | --- | --- |
> | 0.1 0.1 0.8 | 7.47 |
> | 0.1 0.2 0.7 | 10.28 |
> | 0.1 0.3 0.6 | 13.08 |
> | 0.1 0.4 0.5 | 15.89 |
> | 0.2 0.1 0.7 | 10.80 |
> | 0.2 0.2 0.6 | 13.60 |
> | 0.2 0.3 0.5 | 16.41 |
> | 0.2 0.4 0.4 | 19.22 |
> | 0.3 0.1 0.6 | 14.13 |
> | 0.3 0.2 0.5 | 16.93 |
> | **0.3 0.3 0.4** | **19.74** |
> | 0.4 0.1 0.5 | 17.46 |
>
> The results indicate that while there are numerical differences in CACC scores across different weight combinations, the overall trend remains consistent. The default weights adopted in this study (0.3 / 0.3 / 0.4) achieved the highest comprehensive score among all settings and most clearly distinguished the difficulty gap between Q4 and Q1/Q2. Thus, this design not only aligns with the task dependency structure (where Q4 builds on Q1/Q2) but also empirically delivers the most stable and discriminative evaluation outcomes. Overall, the current hyperparameter selection is reasonable.
>
> **Q4-2**: While motivated, this mixes upstream recognition errors with conflict reasoning, and the ablation shows only marginal gains when injecting Q1/Q2 answers directly into the Q4 prompt.
>
> **A4-2**: We use the upstream recognition results as input for Q4 to evaluate the true performance of current MLLMs. To address your concern, we conducted supplementary experiments using the gold labels of Q1/Q2 as input for Q4, and the results are presented below:
>
> | Method | CACC |
> | --- | --- |
> | Base Prompt | 19.231 |
> | + gold Q1 Answer | 20.440 |
> | + gold Q2 Answer | 20.437 |
> | + gold Q1&Q2 Answer | 21.664 |
>
> The experimental results indicate that incorporating Q1/Q2 gold labels significantly impacts model performance. This finding aligns with the observation in Q4-1, confirming a moderate correlation between Q4 and Q1/Q2 (with correlation coefficients of 0.56 and 0.51, respectively). Additionally, it demonstrates that current MLLMs still have substantial room for improvement in cultural conflict-related tasks.

---

> ### Author Response · Authors · 2025-11-20
> **Response to Reviewer oEmH (Part 4)**
>
> **Q5**: How were stereotypes and sensitive portrayals handled during generation and annotation? Any expert review (region-specific annotators) or exclusion lists? A brief ethics appendix would help, given synthetic cultural content concerns.
>
> **A5**: As detailed in Lines 991–994 of Appendix E, expert review was integrated into steps of the text-to-image generation process. This rigorous procedure was implemented to minimize the inclusion of stereotypes and sensitive portrayals as much as possible.
>
> In fact, stereotypes and sensitive portrayals are still products inherent to the specified cultures. Researchers aim for current MLLMs to understand such content, thereby avoiding generating it in real-world scenarios.
>
> **Q6**: For Q4, do you have independent gold conflict triples (culture, object, contradiction) for a subset, rather than relying on model-conditioned answers? If not, could you release human-vetted conflict graphs to decouple recognition errors from reasoning?
>
> **A6**: In fact, we do have gold conflict triples—this subset is derived from the text-construction phase (Step 1 in Figure 3). To ensure data quality and avoid relying on model-conditioned answers, we incorporated a manual inspection step during the construction of these gold conflict triples. Further details regarding this process have been added to Lines 227–228 of the revised manuscript.
>
> **Q7-1**: The human evaluation study needs further elaboration. You bucket difficulty by the number of cultures per image, then sample 100 items per tier. Could you add inter-rater agreement and per-tier time-to-answer?
>
> **A7-1**: To address your concern, we have supplemented the experimental results for inter-rater agreement and per-tier time-to-answer, as follows:
>
> | **Difficulty Tier** | **Inter-rater Agreement** | **Per-tier time-to-answer** |
> | --- | --- | --- |
> | Easy | 95% | 27.01 |
> | Medium | 91% | 52.92 |
> | Hard | 93% | 60.61 |
>
> The results demonstrate high consistency among annotators, while the average response time indicates that the task remains challenging even for human participants. Thank you for your valuable comment—we have updated the corresponding information in Section 5.6.
>
> **Q7-2**: Also, show model vs. human delta broken down by specific culture sets (e.g., lesser-known vs. globally familiar).
>
> **A7-2**: In practice, we have analyzed the model’s performance distribution across different cultures in Section 5.5, where the results indicate that models achieve better performance on globally familiar cultures compared to lesser-known ones.
>
> To further address your question for human analysis (Section 5.6) , we randomly sampled 50 questions each from lesser-known cultures (Iceland, Somalia, Nigeria) and globally familiar cultures (China, Japan, the United States) in the C3B dataset, and invited human participants to answer them. The results for Q1 are presented as follows:
>
> |  | **ACC(lesser-known)** | **ACC(globally familiar)** |
> | --- | --- | --- |
> | Human Analysis | 62% | 74% |
>
> The experimental results demonstrate that models and humans exhibit the same trend across lesser-known and globally familiar cultures: both perform better on globally familiar cultural contexts.
>
> **Q8**: Background Culture Identification and Culture-aware Object Detection: How are gold labels established? Please describe annotator guidelines, the source of culture/object lists, and any expert review. Include inter-annotator agreement (e.g., Cohen’s κ) and examples of borderline cases. Explicitly discuss bias mitigation (e.g., stereotype checks, exclusion lists, region-specific reviewers).
>
> **A8**: In fact, the construction method of the gold labels has already been described in Lines 257–265 of the manuscript, which is primarily determined through manual annotation. We have supplemented additional details about the annotation guidelines in Appendix E. Specifically, we imposed constraints on and revised the annotation results through team discussions and manual inspections. our inter-annotator agreement score (Cohen’s κ) is **76.8%**. As described in Lines 1004–1008, we ensured the accuracy of annotations by cross-referencing credible online sources. Borderline cases are solved by native culture-expert review and their culture-specific knowledge. Leveraging these sources also allowed us to mitigate potential biases to the best of our ability.

---

> ### Author Response · Authors · 2025-11-20
> **Response to Reviewer oEmH (Part 5)**
>
> **Q9-1**: How do you define cultural conflict? Provide an explanation and decision rules with concrete examples from the annotation/internal validation process and guidelines.
>
> **A9-1**: In our work, cultural conflict is defined as follows: a scenario where an image contains two or more distinct cultures simultaneously is regarded as a cultural conflict. Importantly, the "cultural conflict" we refer to does not denote aggressive conflict; instead, it leans toward the meaning of "cultural co-occurrence." This definition is designed to evaluate MLLMs’ capabilities in multicultural scenarios.
>
> We have supplemented a more detailed explanation of the definition and construction method of cultural conflict in Lines 204–206 of the revised manuscript.
>
> **Q9-2**: Clarify whether conflicts are culture–culture, culture–object, or object–object, and how multi-culture scenes are handled.
>
> **A9-2**: Our culture is built on entities, and conflicts between entities are culture-culture. If there are two entities in the same cultural circle in the picture, it does not count as a conflict between the two.
>
> **Q10-1**: Clarify whether the Generation@Culture task uses OCR, classic machine translation, or general-purpose generative LLMs for translation.
>
> **A10-1**: No OCR was used, because the source sentences for translation are the gold source sentences inherently provided in the comic dataset. For each model, the input to the Generation@Culture task consists of the comic images themselves and the gold source sentences, with the MLLM outputting the text translation results.
>
> **Q10-2**: Is the input the manga image or the transcription of the text presented in the image?
>
> **A10-2**: The input to Generation@Culture consists of the comic images themselves and the source sentences.
>
> **Q11-1**: Low BLEU: disentangle error sources. Are low scores primarily due to translation quality or OCR extraction errors?
>
> **A11-1**: Since no OCR was used in the Generation@Culture task, the low BLEU scores can be attributed to the inherent limitations of MLLMs’ multilingual capabilities.
>
> **Q11-2**: If OCR is used on Manga109, I would recommend first reporting OCR accuracy or manual correction rates. State whether you have ground-truth transcriptions for source text from Manga109; if yes, use them to isolate translation quality from text extraction noise.
>
> **A11-2**: Since OCR was not used in our original setup, the scenario you described does not apply. To address your query, we modified the workflow to first perform OCR [1] on Manga109, followed by translation. The experimental results are presented below (BLEU scores for the JA-DE, JA-RU, JA-ES, and JA-TH tasks, respectively):
>
> |  | **OCR** | **Real** | **OCR** | **Real** | **OCR** | **Real** | **OCR** | **Real** |
> | --- | --- | --- | --- | --- | --- | --- | --- | --- |
> | Qwen2.5-VL | 0.92 | 12.0 | 0.97 | 8.74 | - | - | - | - |
> | LLaVA1.5-7B | 0.64 | 2.74 | 0.24 | 1.39 | 0.62 | 4.23 | 0.32 | 1.13 |
> | Llama3.2 | 0.33 | 4.27 | 0.22 | 1.70 | 0.45 | 5.73 | 2.73 | 0.99 |
>
> The results demonstrate that OCR indeed impacts the final translation accuracy.
>
> [1] https://github.com/PaddlePaddle/PaddleOCR, **star 64.4k**
>
> **Q12**: Grammar mistakes/Wording fixes
>
> **A12**: We have incorporated these revisions in the updated version of the manuscript. Thank you for your comment.

---

> ### Author Response · Authors · 2025-11-20
> **Response to Reviewer oEmH (Part 6)**
>
> **Q13**: Given that culture-related benchmarks are still relatively scarce, I recommend expanding the Related Work 2.2 section to include recent datasets that explicitly address cultural diversity in multimodal evaluation. For instance, the VizWiz cultural extension ([Karamolegkou et al., 2024](https://aclanthology.org/2024.hucllm-1.5.pdf)) demonstrates that existing real-world benchmarks, such as VizWiz, encode cultural concepts that were not taken into account in the original crowdsourced annotations. Thus, it introduces a culture-centric evaluation subset in which some images are annotated with more than one culture (e.g., a US brand package of matcha tea, a French wine packaged by Tesco in the UK, containers of Asian food from an Australian meal delivery company, etc.). This work is relevant to your focus on cross-cultural visual understanding and includes images taken in real-world settings by people who are blind that reference more than one culture. Another relevant work that seems to be missing is GlobalRG ([Bhatia et al., 2024](https://aclanthology.org/2024.emnlp-main.385.pdf)), which introduces a multicultural benchmark and two novel tasks—Retrieval across Universals and Cultural Visual Grounding—designed to assess vision-language models' performance across culturally diverse and culture-specific concepts. Including this paper would strengthen the discussion on cultural inclusivity and dataset diversity in VLM evaluation. In addition, ALM-bench ([Vayani et al., 2025](https://arxiv.org/pdf/2411.16508)) represents the largest-scale effort to date to evaluate multimodal models across 100 languages and 13 cultural dimensions, explicitly measuring cultural and linguistic inclusivity. Its scope and methodological framing make it a strong point of comparison for your benchmark. You may also consider referencing [Zhong et al., 2025](https://arxiv.org/html/2510.05931v2), which introduces a complementary perspective on multimodal cultural reasoning and could help position C3B more clearly within the rapidly growing ecosystem of cross-cultural LMM evaluation.
>
> **A13**: Thank you for your suggestion. We have analyzed and discussed these papers in the Related Work section of our manuscript.
>
> **Q14**: Lines 270–280: I would recommend adding a quantitative summary of that inspection (e.g., number reviewed, acceptance rate, inter-rater agreement).
>
> **A14**: We have supplemented additional details about the annotation process in the appendix. Specifically, the annotation for Extraction@Culture and Conflict@Culture is conducted on a per-image basis, with each image corresponding to four subtasks. As provided in Lines 996–997 and Lines 1001–1002, we report two key metrics for quality control: the correction rate from manual inspection (**73.8%**) and the inter-annotator agreement score (**76.8%**).
>
> **Q15**: For instruction-following issues (e.g., “stubbornness” in LLaVA-1.5-7B), this is a prompt compliance problem rather than cultural understanding. Consider simple schema validation (e.g., JSON schema with pydantic) to enforce output format and reduce evaluation noise; note it as an implementation detail, not a scientific claim.
>
> **A15**: We hold the same view regarding the stubbornness phenomenon. Notably, among all tested models, LLaVA-1.5-7B was the sole one that displayed this stubborn trait after we unified the prompt settings. To address the question of whether specific implementation details could mitigate this issue for LLaVA-1.5-7B, we carried out a targeted experiment.
>
> | **Model** | **Q4** |
> | --- | --- |
> | LLaVA-1.5-7B | 0.00 |
> | LLaVA-1.5-7B /w JSON schema with pydantic | 0.00 |
>
> Despite the use of a JSON schema with Pydantic, this issue remains unresolved—indicating that the root cause lies with the model itself.

---

> ### Author Response · Authors · 2025-11-20
> **Response to Reviewer oEmH (Part 7)**
>
> **Q16**: For cultural density measures (CDPI/CBI/CAD), add brief intuition + formulae in main text and validate that higher density correlates with lower model accuracy on held-out items.
>
> **A16**: In practice, we have added the brief intuition and formulae for CDPI, CBI, and CAD in Section 5.4. Additionally, the difficulty stratification presented in Section 5.6 is directly based on these three metrics, demonstrating that higher cultural density (as quantified by CDPI/CBI/CAD) correlates with lower model accuracy. To further validate this correlation, we have supplemented Table 7 with the specific numerical values of CDPI, CBI, and CAD. The experimental results are presented below:
>
> |  | **CDPI** | **CBI** | **CAD** | **ACC-Q1** | **ACC-Q2** | **ACC-Q3** | **CACC-Q4** |
> | --- | --- | --- | --- | --- | --- | --- | --- |
> | Easy | 1.73 | 77.85 | 9.50 | 48.0 | 28.8 | 50.5 | 23.8 |
> | Medium | 8.65 | 614.15 | 53.20 | 32.6 | 27.1 | 54.0 | 18.6 |
> | Hard | 11.85 | 793.95 | 71.88 | 25.7 | 25.5 | 41.7 | 15.9 |
>
> The results indicate that the three metrics exhibit a monotonically increasing trend across the three difficulty levels, confirming the rationality of our stratification criteria. Concurrently, model performance consistently decreases as difficulty increases, which aligns with our initial hypothesis. Thank you for your comment.
>
> **Q17-1**: If you claim “progressive difficulty,” define how difficulty is constructed and validated (e.g., number of cultures per image, ambiguity, required cross-culture reasoning) and show that model performance monotonically decreases with the proposed difficulty index (or explain deviations).
>
> **A17-1**:  We have designed the progressive difficulty based on task types. Specifically, Extraction@Culture and Conflict@Culture focus on understanding, while Generation@Culture centers on generating—and generating tasks are generally more challenging than understanding tasks in NLP research [1-5].
>
> Actually, our experimental results show a clear performance gradient: the model achieves the highest ACC on Extraction@Culture, followed by Conflict@Culture, with the lowest performance on Generation@Culture. Additionally, we conducted ablation studies on Extraction@Culture and Conflict@Culture in Section 5.3, confirming the monotonically decrease.
>
> In summary, the progressive difficulty of our task design is both reasonable and logically rigorous. Thank you for your valuable comment.
>
> [1] Multimodal attention with image text spatial relationship for ocr-based image captioning. ACM MM 2020.
>
> [2] Sieve: Multimodal dataset pruning using image captioning models. CVPR 2024.
>
> [3] Polos: Multimodal metric learning from human feedback for image captioning. CVPR 2024.
>
> [4] Survey of Low-Resource Machine Translation. Computational Linguistics 48.3 (2022).
>
> [5] Bridging Sparse Domain Semantics via an Asymmetric Siamese Framework with Virtual Anchor Guidance for Domain-Specific Multimodal Translation. Artificial Intelligence (2025): 104443.
>
> **Q17-2**: You should also clarify that by progressive difficulty, you also include the dimension of how many cultural objects are contained in one image (as it is only briefly mentioned in the human-model analysis).
>
> **A17-2**: In fact, we have conducted a comprehensive analysis of the number of cultural objects using CDPI, CBI, and CAD. Actually, we defined task difficulty based on the number of cultural objects **ONLY** in Section 5.6.
>
> To address your concern, we calculated the CDPI, CBI, and CAD values for images across different difficulty levels. The results indicate that these three metrics exhibit a monotonically increasing trend with the ascending difficulty levels, providing empirical evidence for the rationality of our classification criteria.
>
> |  | **CDPI** | **CBI** | **CAD** |
> | --- | --- | --- | --- |
> | Easy | 1.73 | 77.85 | 9.50 |
> | Medium | 8.65 | 614.15 | 53.20 |
> | Hard | 11.85 | 793.95 | 71.88 |

---

> ### Author Response · Authors · 2025-11-20
> **Response to Reviewer oEmH (Part 8)**
>
> **Q18**: To compensate for the limitations of synthetic data and artificial tasks, you could draw inspiration from [Evaluating Multimodal Language Models as Visual Assistants for Visually Impaired Users](https://aclanthology.org/2025.acl-long.1260/) (Karamolegkou et al., ACL 2025). That work grounds multimodal evaluation in real-world needs, such as question answering and image captioning for accessibility. You should acknowledge it in related works, since MLMMs are now being used in the real world. Incorporating similarly practical, human-centered tasks would make C3B more ecologically valid and demonstrate the benchmark’s relevance beyond synthetic cultural scenarios.
>
> **A18**: We are pleased that you referenced the paper *"Evaluating Multimodal Language Models as Visual Assistants for Visually Impaired Users"*. This work designs tasks including "Braille-to-Text Translation" and "Cross-script QA"—findings that strongly validate the rationality of our dataset construction and task design. Thank you for this valuable suggestion; we will cite and analyze this paper in the Related Work section.
>
> Regarding human-centered tasks, we fully agree with your insight that such tasks are "more ecologically valid and demonstrate the benchmark’s relevance beyond synthetic cultural scenarios." This aligns with the future direction of AI development, and we plan to incorporate this focus in our future work.
>
> Additionally, comics have long been a real-world medium—even before the widespread use of photography, illustrations and comics served as a means to document reality. Our work is the first to introduce comics as a core theme in this research area, and we believe it will provide insights for future development of comic-centric, human-centered datasets. Thank you for your comment.

---

> ### Author Response · Authors · 2025-11-27
> **Gentle Reminder for Reviewer oEmH**
>
> Dear Reviewer oEmH,
>
> We would like to thank you once again for your review of our submission.
>
> In our prior response, we addressed your concerns point-by-point with comprehensive information, covering aspects such as benchmark design (Q1-1\~Q4-2), details of MT tasks (Q10-1\~Q11-2), human annotation procedures (Q5\~Q9-2, Q14), hyperparameter configurations (Q4-1) and experiment details (Q15\~Q17-2).
>
> To further address your concern about integrating OCR into MT tasks (Q11-2), we have completed all the experiments during this period. The results are as follows: (BLEU scores for the JA-DE, JA-RU, JA-ES, and JA-TH tasks, respectively)
>
> |  | **OCR** | **Real** | **OCR** | **Real** | **OCR** | **Real** | **OCR** | **Real** |
> | --- | --- | --- | --- | --- | --- | --- | --- | --- |
> | Qwen2.5-VL | 0.92 | **12.0** | 0.97 | **8.74** | 1.32 | **14.5** | 1.22 | **9.72** |
> | LLaVA1.5-7B | 0.64 | **2.74** | 0.24 | **1.39** | 0.62 | **4.23** | 0.32 | **1.13** |
> | Llama3.2 | 0.33 | **4.27** | 0.22 | **1.70** | 0.45 | **5.73** | 2.73 | **0.99** |
>
> From the results, we can see that incorporating OCR clearly reduces the model's performance on generation tasks. This further shows that our method does not rely on OCR but translates directly from images and text.
>
> We would be happy to provide further discussion otherwise. Your insights are valuable to us, and we appreciate your time and attention to our work. Look forward to your feedback!
>
> Thank you again for your comments!
>
> Best Regards,
>
> All Authors

---

> ### Comment · Reviewer_oEmH · 2025-11-27
> **Response to Authors**
>
> I really appreciate your detailed responses. After carefully reviewing your clarifications and proposed revisions, I have decided to raise my score, provided that the improvements outlined in your rebuttal are fully incorporated into the camera-ready version.
>
> Some comments from my side:
>
> - **On prompt issues**: I agree that detailed prompts can improve LLM performance; however, my comment was not about length alone. It concerned readability, grammatical correctness, and consistency of task framing, especially in the versions shown in Figures 1–2 of the submission and the Appendix. These issues can affect human annotators, reproducibility, and clarity for readers, not only model performance. You have covered my major concern with your response, and I trust you will revise the paper accordingly. A minor point: “contradict” is a transitive verb, so replacing it with the noun “contradiction” in figures and text would improve clarity; please feel free to clarify if you interpret this differently.
> - **On annotation and validation**: You mention that dataset construction is supported by native culture-expert review, yet the manuscript still does not specify the cultural or geographic background of the annotators involved. Since the benchmark explicitly focuses on cultural nuance across nearly 200 regions, providing more detail on annotator backgrounds, instructions, and guidelines would strengthen transparency, reproducibility, and the credibility of the dataset.
> - **On OCR and performance**: It would be valuable to include an analysis in the appendix on whether OCR extraction quality affects model outputs. This information would be highly useful to other researchers working on multimodal cultural benchmarks.
> - **Ethics statement**: I still believe an explicit ethics or impact statement would be beneficial. This section could discuss potential biases, concerns of relying on synthetic content, cultural sensitivity considerations, and dataset licensing. If you choose not to include definitions of cultural contradiction and cultural conflict in the introduction or related work, this would also be an appropriate place to clearly articulate these concepts. A brief discussion of the differences and limitations between real-world images (**which is currently missing**) and comic-style content could further contextualize and support your design choices.

---

> > ### Author Response · Authors · 2025-11-28
> > **Thanks for your support!**
> >
> > We sincerely appreciate your valuable suggestions!
> >
> > As for questions on p**rompt issues**, we are pleased that our experiments have effectively alleviated your concerns. We fully agree with your insight that prompts should also be readable, and we will update this section in the camera-ready version.
> >
> > As for questions on **annotation and validation**, we have updated the regional information of the annotators (Lines 960–964): the annotators are from Asian countries, so cultural experts were involved in annotating Asian-related content. For other cultural contexts, we have carefully consulted credible online resources to ensure annotation quality. We will further supplement and refine this information in the camera-ready version.
> >
> > As for questions on **OCR and performance**, we are pleased that our experiments have effectively alleviated your concerns! We have incorporated the OCR experiment in Lines 1127–1187 of the manuscript.
> >
> > As for questions on **ethics statement**, after careful discussion, we have added an Ethics Statement (Lines 508–516), further clarifying how we addressed biases, stereotypes, and culturally sensitive content in our work. Regarding the definition of cultural conflict, we have supplemented relevant details in Lines 204–206, which we have also restated in the Ethics Statement. Additionally, the differences between manga and real images are discussed in Lines 66–69.
> >
> > Thank you again for your valuable suggestions!

---

### Official Review · Reviewer_h7gx · 2025-11-01

**Soundness:** 1
**Presentation:** 2
**Contribution:** 2
**Rating:** 2
**Confidence:** 4

**Summary:**

This paper proposes benchmarking MLLM in cultural and multilingual understanding via tasks involving comic panel as the image input. This benchmark were generated with a pipeline of both automated generated data and human in-the-loop for verification, and have 3 sub-tasks, covering various languages depening on the task. Lastly, author benchmarked various models, noting the challenge of the benchmark.

**Strengths:**

The use of comic-panel as a benchmark is interesting and unique. This benchmark is decently sized, multilingual.

**Weaknesses:**

My biggest concern is the decision to use AI-generated images for this particular benchmark. If we hypothesize that current MLLMs are not yet proficient at handling multicultural contexts, then generating cultural images in this way could be harmful and potentially misleading. Anecdotally, the examples in this paper show exaggerated depictions of cultures (see relevant reading: https://arxiv.org/pdf/2407.14779v1).

This issue is further compounded by the lack of human validation for the generated images. Without proper grounding (e.g., visual references), there is no guarantee that the generated cultural images are accurate or appropriate.

The author also does not seem to consider what truly constitutes a culture, beyond surface-level representations such as “Native American” or related objects. While these can serve as proxies for culture, cultural understanding goes much deeper than that. The paper could benefit from a more robust definition of what the author considers a “cultural benchmark,” which would also clarify how the list of cultures, objects, or related proxies was derived.

Figure 5 further illustrates the lack of care in defining culture. The plot is confusing, it includes “Asia” alongside “Indonesia,” “Korea,” and “Japan.” Are these not all Asian cultures? What, then, constitutes “Asian” culture? This connects back to the earlier issue of validation: just as the generated images require validation, the questions themselves should also be validated, ideally by native speakers and individuals familiar with the respective cultures.

Minorly, the author claims that existing cultural VQA datasets are weaker because they use real-world images, implying those are easier. I am not convinced by this reasoning. Real-world images reflect the actual use cases for these models, and there is no guarantee that such tasks are easier.

**Questions:**

-

---

> ### Author Response · Authors · 2025-11-20
> **Response to Reviewer h7gx (Part 1)**
>
> **Q1**: My biggest concern is the decision to use AI-generated images for this particular benchmark. If we hypothesize that current MLLMs are not yet proficient at handling multicultural contexts, then generating cultural images in this way could be harmful and potentially misleading. Anecdotally, the examples in this paper show exaggerated depictions of cultures (see relevant reading: https://arxiv.org/pdf/2407.14779v1).
>
> **A1:** Actually, MLLMs do not generate the comics themselves; they are only used to understand the comics and produce textual responses. The comic images are generated by Text2Image models. Although Text2Image models [1–3] may introduce bias, prior work shows that they are capable of producing culturally accurate content [4–5]. Moreover, the paper (https://arxiv.org/pdf/2407.14779v1) you recommended contains exaggerated depictions of cultures, but the scenes it shows are still within the Indian cultural sphere. In addition, for our Generation@Culture task, we also include comics drawn by real artists, which further mitigates the potential impact of generation bias.
>
> [1] High-Resolution Image Synthesis with Latent Diffusion Models, CVPR, cited 26501.
>
> [2] Hierarchical Text-Conditional Image Generation with CLIP Latents, OpenAI, cited 9248
>
> [3] Photorealistic Text-to-Image Diffusion Models with Deep Language Understanding, Google, cited 8239
>
> [4] Beyond aesthetics: Cultural competence in text-to-image models. NeurIPs 2024
>
> [5] CULTURALFRAMES: Assessing Cultural Expectation Alignment in Text-to-Image Models and Evaluation Metrics, EMNLP 2025
>
> **Q2**: This issue is further compounded by the lack of human validation for the generated images. Without proper grounding (e.g., visual references), there is no guarantee that the generated cultural images are accurate or appropriate.
>
> **A2**: In our paper (Lines 257–290), we describe that the entire pipeline—from annotating image elements, constructing questions, to generating images—involves human annotation and verification. At every stage, human annotators inspect and label the elements within each image to ensure that both the generated images and the constructed questions are accurate and appropriate. To further address your concern, we additionally compute CLIPScore to evaluate whether the generated images align accurately and appropriately with their corresponding text. We also apply the same evaluation procedure to real-world image–based cultural QA datasets [1–3]. The results are as follows:
>
> | **Dataset** | **CLIPScore** |
> | --- | --- |
> | MOSAIC-1.5k [1] | 0.7245 |
> | CapArena [2] | 0.7944 |
> | C3B | 0.7337 |
> | VizWiz-val [3] | 0.7344 |
> | VizWiz-train [3] | 0.7394 |
>
> Our experimental results show that the generated comic images are not weaker than real cultural images in terms of accuracy or appropriateness. In addition, one-third of the C3B images are drawn by real comic artists, making the proportion between generated and real images well balanced. Thank you for your comment.
>
> [1] How Culturally Aware Are Vision-Language Models?. IPAS 2025.
>
> [2] Caparena: Benchmarking and analyzing detailed image captioning in the llm era. arXiv 2025.
>
> [3] VizWiz Grand Challenge: Answering Visual Questions from Blind People. CVPR 2018.

---

> > ### Comment · Reviewer_h7gx · 2025-11-21
> >
> > hank you for your response.
> >
> > Q1 & Q2: I want to clarify that my focus was on AI-generated images instead of one specific VLLM. The mention of VLLM being bad in this aspect was because the technology is still subpar. In what sense can we trust automatically generated images? I believe manual verification only happens in step 7, which I think covers only part of the process, rather than as early as when the image is generated (for example, step 3). Who are the verifiers, and do they have an understanding of the culture they are validating?
> >
> > In CulturalFrames, it shows that models still have room for improvement in cultural alignment, and depending on the culture, the performance gets worse. This is relevant to the need for human verification on the image generation side, with a proper annotation guideline (the exact, verbatim instructions given to the annotators, rather than just a summary) and reported annotator demographics (are they native to the region? Would learning from Wikipedia be enough to understand?). This ensures that the generated image is indeed correct and trustworthy. See my next point.
> >
> > Perhaps we can look at one example: your Figure 6. Many things are wrong here, which raises questions about data quality in general.
> >
> > 1. It is not a kilt, since it looks like pants.
> >
> > Models make seemingly small errors, but they still change the cultural correctness of the image. Some objects are very intricate and similar. For example, Malaysian Batik and Indonesian Batik look similar but have some differences. If the model cannot deal with the difference between pants and a skirt-like garment, how can we assume the model can handle intricate concepts?
> >
> > 2. Why is the man an Australian surfer? Why not from a different nationality? This can be racist or stereotypical.
> >
> > 3. The surfer’s facial features have changed, which raises questions about the overall quality. The 'kilt (but not really) also worn by the supposed 'Arabic' man.
> >
> > 4. The supposed 'Arabic' man’s headwear has changed shape. Similar to point 1, each culture has small differences that the model may not be able to capture.
> > For example, I suggest you look up a Saudi Thobe versus a UAE Kandura. They might look similar, but they have their differences. Can you guarantee your model or annotator knows this? The change in the man’s headwear might suggest otherwise. Similarly, it could resemble Pakistani or Afghan headwear (which are not Arab, by the way).
> >
> > 5. How is that place means in Brazil? rainforest hut is not exclusive to Brazil.
> >
> > This also connects to your next comment on the categorization between specific countries and generic regions like 'Asia.' The borders are vague and hard to define, and you have not described in detail how the decisions were made. Given how intricate these distinctions are, I suggest providing clearer definitions and annotation guidelines to ensure the data is correct, high-quality, and not stereotypical, as it currently seems to be.
> >
> > My point still stands that I do not think AI-generated data is suitable for your purpose.

---

> > > ### Author Response · Authors · 2025-11-26
> > > **Response to Reviewer h7gx (Part 1)**
> > >
> > > **Q1-1**: Q1 & Q2: I want to clarify that my focus was on AI-generated images instead of one specific VLLM. The mention of VLLM being bad in this aspect was because the technology is still subpar. In what sense can we trust automatically generated images?
> > >
> > > **A1-1**: We don’t think "the technology is still subpar." Contemporary AI-generated images can already achieve photorealism, with some even becoming indistinguishable from real images in terms of authenticity [1-4].
> > >
> > > Regarding the question of "in what sense we can trust generated images," AI-generated imagery has undergone nearly a decade of research [5-9], leading to the development of a mature automated metrics. Metrics such as CLIPScore [10], trained on extensive **real** image-text pairs, directly reflect the authenticity and semantic consistency of generated images, enabling quantitative verification of their reliability.
> > >
> > > To further address this concern, we computed CLIPScore for both C3B and a cultural image-text datasets composed of real images. Our results demonstrate that C3B’s CLIPScore is on par with that of the real-image dataset and even higher than certain subsets. These findings thus validate the quality of the generated images in C3B.
> > >
> > > | **Dataset** | **CLIPScore** |
> > > | --- | --- |
> > > | MOSAIC-1.5k | 0.7245 |
> > > | CapArena | 0.7944 |
> > > | C3B | 0.7337 |
> > > | VizWiz-val | 0.7344 |
> > > | VizWiz-train | 0.7394 |
> > >
> > > To more fully address the concern regarding "trust in generated images," we designed an experiment to verify the cultural reliability of the generated images. Our hypothesis is as follows: If the generated images genuinely encode culture-related information, models fine-tuned on these images should achieve improved performance on cultural tasks involving real images; conversely, if the generated images lack cultural correctness, the model’s performance will significantly degrade.
> > > Guided by this logic, we fine-tuned Qwen-2.5-VL-7B and Llama3.2-8B using Q1 data from C3B and evaluated the model on CVQA (a cultural dataset composed entirely of real images). The results demonstrate a significant improvement in model accuracy. This performance gain confirms that the generated images indeed provide valid cultural knowledge rather than introducing biases or noise.
> > >
> > > | **Method** | **ACC** |
> > > | --- | --- |
> > > | Qwen-2.5-VL w/o C3B | 19.6 |
> > > | Qwen-2.5-VL w/ C3B | **27.1** |
> > > | Llama3.2 w/o C3B | 24.1 |
> > > | Llama3.2 w/C 3B | **28.9** |
> > >
> > > Thus, this experiment offers direct, quantitative, and positive evidence for the cultural reliability of the generated images. Thank you for your valuable comment—we will update this result in the next version.
> > >
> > > [1] Gemini 3 Pro: https://blog.google/technology/ai/nano-banana-pro/
> > >
> > > [2] Zero-shot detection of ai-generated images. ECCV 2024.
> > >
> > > [3] Secret Lies in Color: Enhancing AI-Generated Images Detection with Color Distribution Analysis. CVPR 2025.
> > >
> > > [4] Chu, Beilin, et al. "Fire: Robust detection of diffusion-generated images via frequency-guided reconstruction error. CVPR 2025.
> > >
> > > [5] Where did i come from? origin attribution of ai-generated images. NeurIPS 2023.
> > >
> > > [6] Wildfake: A large-scale and hierarchical dataset for ai-generated images detection. AAAI 2025.
> > >
> > > [7] Organic or diffused: Can we distinguish human art from ai-generated images. SIGSAC 2024.
> > >
> > > [8] Conditional image generation with pixelcnn decoders. NeurIPS 2016.
> > >
> > > [9] Image generation from scene graphs. CVPR 2018.
> > >
> > > [10] Clipscore: A reference-free evaluation metric for image captioning. ACL 2021.
> > >
> > > **Q1-2**: I believe manual verification only happens in step 7, which I think covers only part of the process, rather than as early as when the image is generated (for example, step 3).
> > >
> > > **A1-2**: Manual verification happens at **multiple stages**, not only at Step 7. In Figure 3, the human-shaped marker indicates manual verification. Our pipeline contains four layers of human verification: (1) Prompt verification before image generation. As described in Lines 225–229, all text prompts are manually checked to filter out culturally incorrect or sensitive content before any image is generated. (2) Cultural element annotation (Step 3). Lines 962–965 describe the manual creation of the *Culture List* and *Cultural Objects List*. Annotators verify all cultural elements appearing in each image and correct any mismatch. This ensures that cultural information is accurate **at the earliest stage where the image is inspected**. (3) Question construction (Steps 4–5). Because question writing depends on the annotated cultural elements, every question is again reviewed by annotators (Lines 261–265). This acts as the second-level consistency check between text and image. (4) Question refinement and ground-truth checking (Steps 7–9). Refinement is also fully manual (Lines 274–278), and all ground-truth translations in the Generation@Culture task are validated by humans.

---

> > > ### Author Response · Authors · 2025-11-26
> > > **Response to Reviewer h7gx (Part 2)**
> > >
> > > **Q1-3**: Who are the verifiers, and do they have an understanding of the culture they are validating?
> > >
> > > **A1-3**: All annotations were performed by three graduate students and two undergraduate students. Among the five annotators, three are graduate students engaged in culture-related research. We have supplemented the detailed information of the annotators in Appendix E. Because the annotators first used credible external online resources (English Wikipedia) to ensure the correctness of textual information, and then filtered AI-generated images using the images provided by Wikipedia, the cultural quality of both text and images was ensured.
> > >
> > > **Q2-1**: In CulturalFrames, it shows that models still have room for improvement in cultural alignment, and depending on the culture, the performance gets worse.
> > >
> > > **A2-1**: In fact, our experimental results show that the performance difference stems mainly from variations in task difficulty rather than the low quality of the created images. As seen in Table 3, several models achieved over 50% accuracy. If the image quality were poor, the maximum accuracy would be much lower.
> > > Thus, the drop in MLLMs’ performance reflects increased task complexity, not a lack of cultural correctness in the created images.
> > >
> > > **Q2-2**: This is relevant to the need for human verification on the image generation side, with a proper annotation **guideline** (the exact, verbatim instructions given to the annotators, rather than just a summary) and reported annotator demographics (are they native to the region? Would learning from Wikipedia be enough to understand?). This ensures that the generated image is indeed correct and trustworthy. See my next point.
> > >
> > > **A2-2**: Manual annotation is integrated throughout the entire pipeline. Three of our five annotators have culture-related research backgrounds, and detailed annotation guidelines are provided in Lines 1008–1025. As stated in Lines 225–229, all prompts undergo manual cultural verification before image generation. The culture list, cultural objects list, and all questions are also manually curated. Together, these dual constraints on images and questions ensure the cultural accuracy of our benchmark.
> > >
> > > **Q3**: Perhaps we can look at one example: your Figure 6. Many things are wrong here, which raises questions about data quality in general.
> > >
> > > 1. It is not a kilt, since it looks like pants.If the model cannot deal with the difference between pants and a skirt-like garment, how can we assume the model can handle intricate concepts?
> > >
> > > **A3**: For this question, the garment is treated as a kilt. Since the task uses multiple-choice and all options include “kilt,” this detail does not affect the answer. It reflects how we align options, images, and questions to maintain benchmark quality.
> > >
> > > Furthermore, comics created by professional artists do not aim for perfect realism. They often simplify or exaggerate real-life elements while preserving their recognizable cultural meaning. For example, Disney’s animated film Mulan does not reproduce Chinese culture with complete accuracy, yet audiences in both China and the West can still immediately identify it as reflecting Chinese culture. Therefore, our benchmark aims not to mimic real scenes but to test whether MLLMs can recognize and generate cultural cues.
> > >
> > > [1] "Hybridising the cultural identity of Mulan from a Chinese ballad to American films." *Asian Journal of Social Science* 50.2 (2022): 130-136.
> > >
> > > **Q4-1**: Models make seemingly small errors, but they still change the cultural correctness of the image. Some objects are very intricate and similar. For example, Malaysian Batik and Indonesian Batik look similar but have some differences.
> > >
> > > **A4-1**: The same logic as in Q3 applies. The Batik option is intentionally designed as a distractor in the multiple-choice setting, so it is not part of the correct answer and does not affect the cultural validity of the question. Therefore, we design the option–answer–question annotation process carefully to prevent small errors from affecting the quality of the benchmark.
> > >
> > > **Q4-2**: Why is the man an Australian surfer? Why not from a different nationality? This can be racist or stereotypical.
> > >
> > > **A4-2**: The “Australian surfer” in the options comes from the prompt used to generate the image. In fact, there is a dedicated Wikipedia entry titled “Surfing in Australia” [1]. During manual annotation, we constructed accurate prompts based on existing cultural knowledge. Therefore, including “Australian surfer” in both the options and the image is not racist or stereotypical.
> > >
> > > [1] https://en.wikipedia.org/wiki/Surfing_in_Australia

---

> > > ### Author Response · Authors · 2025-11-26
> > > **Response to Reviewer h7gx (Part 3)**
> > >
> > > **Q4-3**: The surfer’s facial features have changed, which raises questions about the overall quality. The 'kilt (but not really) also worn by the supposed 'Arabic' man.
> > >
> > > **A4-3**: Yes, the surfer only appears in the first frame because the surfboard provides a unique  visual cue. Therefore, the other characters are not surfers. In addition, we did not bind the concept of “surfer” to “kilt,” so it is expected that the kilt may appear on characters from other cultural backgrounds. Finally, through our construction process, we ensure that the correct answer remains uniquely identifiable.
> > >
> > > **Q4-4**: The supposed 'Arabic' man’s headwear has changed shape. Similar to point 1, each culture has small differences that the model may not be able to capture. For example, I suggest you look up a Saudi Thobe versus a UAE Kandura. They might look similar, but they have their differences. Can you guarantee your model or annotator knows this? The change in the man’s headwear might suggest otherwise. Similarly, it could resemble Pakistani or Afghan headwear (which are not Arab, by the way).
> > >
> > > **A4-4**: We appreciate your attention to "cultural detail differences." Our partly main purpose is to enable models to identify **primary cultural cues,** rather than distinguishing between fine-grained differences between several countries or ethnic groups. Items such as the Saudi Thobe and the UAE Kandura do differ regionally, but these belong to intra-region variation. This response also answers the question raised in Q4-1(in **Response to Reviewer h7gx (Part 2)**), that certain labels focus on broader cultural concepts rather than highly detailed distinctions.
> > >
> > > Regarding headwear related to Pakistani and Afghan cultures (such as the Pakol or Karakul hat), we used a manual annotation workflow with external reference materials to remove any images that conflicted with Arab cultural elements, as stated in Appendix E. Therefore, the headwear shown in Figure 6 does not involve cross-cultural mislabeling.
> > >
> > > **Q4-5**: How is that place means in Brazil? rainforest hut is not exclusive to Brazil.
> > >
> > > **A4-5**: Yes. Based on the cue “rainforest hut,” the only valid options are Brazil and Cambodia. We then used information from Wikipedia to annotate and verify the data:
> > >
> > > - **Brazilian** rainforest huts are primarily concentrated in the Amazon Rainforest region, typically near the city of Manaus or along the Amazon River and its tributaries. They range in form from cozy wooden cabins to luxury eco-lodges—some are floating huts (e.g., [Uakari Lodge](https://www.google.com/viewer/place?mid=/g/11_qdw6p_&sa=X&ved=2ahUKEwjs3OeSroeRAxVVi68BHfGyIG0QqdYPegYIAQgFEAM)) while others are built on land.
> > > - **Cambodian** rainforest huts are mainly located within national parks and protected areas such as the Cardamom Mountains—one of Southeast Asia’s last remaining intact large rainforest regions. Accommodations here primarily take the form of tented camps, such as Shinta Mani Wild and Cardamom Tented Camp.
> > >
> > > Based on this, the most reasonable answer is Brazil. This example demonstrates that we conducted thorough design and manual verification to ensure the overall quality.
> > >
> > > This also directly addresses your question: "Would learning from Wikipedia be enough to understand?" All the detailed knowledge presented above is sourced from Wikipedia.
> > >
> > > **Q5-1**: This also connects to your next comment on the categorization between specific countries and generic regions like 'Asia.' The borders are vague and hard to define, and you have not described in detail how the decisions were made.
> > >
> > > **A5-1**: We have addressed the questions related to Figure 5 in Q4-1 and Q4-4. As emphasized earlier, our partly main purpose is to enable models to identify **primary cultural cues**, rather than distinguishing between fine-grained variants within a country or ethnic group.
> > >
> > > For instance, if "mooncake" appears in a question, since they are consumed in multiple Asian countries, selecting "Asia" as the answer is more appropriate. This approach aligns with our purpose.
> > >
> > > **Q5-2**: Given how intricate these distinctions are, I suggest providing clearer definitions and annotation guidelines to ensure the data is correct, high-quality, and not stereotypical, as it currently seems to be.
> > >
> > > **A5-2**: In our earlier replies to Q1-3 and Q2-2, we explained the definitions, guidelines, and human annotation process in detail. Beyond human evaluation, the CLIPScore results and the finetuning experiments in Q1-1 show that the data is correct, high-quality, and not stereotypical.

---

> > > ### Author Response · Authors · 2025-11-26
> > > **Response to Reviewer h7gx (Part 4)**
> > >
> > > **Q6**: My point still stands that I do not think AI-generated data is suitable for your purpose.
> > >
> > > **A6**: We argue that AI-generated images are suitable for our purpose, with three key justifications centered on **controllability**, **risk isolation**, and **validation mechanisms**:
> > >
> > > **(1) Real-world images cannot meet our requirement for "multicultural in one image":** Our benchmark requires controlled combinations of multiple cultural elements, which real images cannot provide because they usually contain only one culture. AI-generated images let us specify cultural objects precisely. We use AI-generated comics because they creatively reshape reality while keeping cultural cues recognizable, similar to the Mulan example in A3.
> > >
> > > **(2) AI-generated images account for 2/3 of the total dataset:** We implemented a rational task division in our benchmark design. Tasks *Extraction@Culture and Conflict@Culture* use AI-generated images to evaluate models’ ability to "understand cultural cues". Task *Generation@Culture* use human-drawn real comics, preventing the potential biases from AI-generated images. Thus, any potential concerns regarding AI-generated images are confined to recognition-based tasks and do not affect the evaluation of generation tasks.
> > >
> > > **(3) The cultural correctness of AI-generated images has been validated through both manual and automated mechanisms:** We conducted multi-stage manual checks (see Q1-2 and Lines 225–229, 261–265, 962–965 for details) and used CLIPScore to confirm that C3B’s visual quality matches or exceeds real cultural datasets. More importantly, fine-tuning on C3B significantly improves CVQA accuracy, which would not happen if the images lacked cultural correctness. This shows that AI-generated images indeed encode reliable cultural knowledge.

---

> ### Author Response · Authors · 2025-11-20
> **Response to Reviewer h7gx (Part 2)**
>
> **Q3-1**: The author also does not seem to consider what truly constitutes a culture, beyond surface-level representations such as “Native American” or related objects. While these can serve as proxies for culture, cultural understanding goes much deeper than that.
>
> **A3-1**: Our work is well situated within the scope of multimodal cultural benchmarks, as current mainstream benchmarks primarily focus on knowledge-centric cultural scenarios [1–5]. Compared with previous cultural benchmarks, we first extend cultural data from real-world photos to the new domain of comics. Second, we construct a set of evaluation tasks covering multiple dimensions, ranging from cultural understanding to cultural generation. Third, we evaluate leading MLLMs and identify several open research challenges. Based on these contributions, Reviewer 6ZTX acknowledges the novelty and depth of our work.
>
> We also acknowledge that exploring datasets with deeper cultural definitions is a meaningful future direction. In addition, we agree that “cultural understanding goes much deeper than proxies for culture,” and research on culture should indeed address these deeper aspects. We will continue to work toward this.
>
> [1] CVQA: Culturally-diverse Multilingual Visual Question Answering Benchmark. NeurIPS 2024.
>
> [2] Benchmarking Vision Language Models for Cultural Understanding. EMNLP. 2024.
>
> [3] CULTURALFRAMES: Assessing Cultural Expectation Alignment in Text-to-Image Models and Evaluation Metrics, EMNLP 2025
>
> [4] BLEND: a benchmark for llms on everyday knowledge in diverse cultures and languages. NeurIPS 2024.
>
> [5] CARE: aligning language models for regional cultural awareness. EMNLP 2025.
>
> **Q3-2**: The paper could benefit from a more robust definition of what the author considers a “cultural benchmark,” which would also clarify how the list of cultures, objects, or related proxies was derived.
>
> **A3-2**: In terms of cultural understanding, C3B evaluates MLLMs’ cultural competence by assessing their grasp of cultural objects, cultural proxies, and the conflicts between them within images.
>
> In terms of cultural generation, C3B measures MLLMs’ ability to generate culturally appropriate content through multilingual and cross-cultural translation tasks.
>
> Regarding the definition and curation of the list of cultures, cultural objects, or related proxies, detailed explanations are provided in Lines 258–261 and Figure 3 of the manuscript:
>
> ```jsx
> For the task Extraction@Culture, we first manually compiled two lists: one containing all distinct cultures present in the comic pages, and the other listing all culturally representative objects.
> ```
>
> **Q4-1**: Figure 5 further illustrates the lack of care in defining culture. The plot is confusing, it includes “Asia” alongside “Indonesia,” “Korea,” and “Japan.” Are these not all Asian cultures? What, then, constitutes “Asian” culture?
>
> **A4-1**: This is because our labeling adopts a granularity-based classification approach. Specifically, if a cultural object depicted in an image can be explicitly attributed to a particular country, it is categorized at the national level; if a cultural object is shared across a broader regional scope, it is classified at the regional level.
>
> To further address your concern, we randomly sampled 50 images from the "Asia"-labeled dataset in Figure 5. These images were then used to evaluate responses to Question 1 (Q1) via three approaches: manual annotation, GPT-4o, and Qwen3-VL-235B-A22B. The experimental results are presented as follows:
>
> |  | **Percentage of answers labeled “Asia”** |
> | --- | --- |
> | Human | 92% |
> | GPT-4o | 80% |
> | Qwen3-VL-235B-A22B | 84% |
>
> Our experimental results indicate that these images themselves are inherently prone to being classified into coarse-grained categories. This finding thus validates the rationality and rigor of our labeling scheme.
>
> **Q4-2**: This connects back to the earlier issue of validation: just as the generated images require validation, the questions themselves should also be validated, ideally by native speakers and individuals familiar with the respective cultures.
>
> **A4-2**: Our overall dataset construction is grounded in contributions from native experts and their domain-specific knowledge. Consequently, cultural knowledge errors and biases have been avoided even during the question validation phase. Detailed descriptions of this process are provided in Lines 1004–1008 of the manuscript. Thank you for your valuable comment.

---

> ### Author Response · Authors · 2025-11-20
> **Response to Reviewer h7gx (Part 3)**
>
> **Q5**: Minorly, the author claims that existing cultural VQA datasets are weaker because they use real-world images, implying those are easier. I am not convinced by this reasoning. Real-world images reflect the actual use cases for these models, and there is no guarantee that such tasks are easier.
>
> **A5**: This appears to be a misunderstanding of our work. As noted in Lines 57–59, real-world images in existing studies contain only one cultural element, making their tasks single-label classification. In contrast, the images in our work include multiple cultural elements, resulting in a multi-label classification task—which is more challenging in terms of comprehension requirements.
>
> Additionally, existing cultural VQA datasets focus solely on comprehension tasks. Our work, however, incorporates a generation task on top of comprehension to further evaluate MLLMs’ ability to generate culturally relevant content.
>
> In summary, our work presents greater challenges compared to existing cultural VQA datasets, both in terms of task design and evaluation dimensions.

---

### Meta-Review · Area_Chair_pUfk · 2025-12-31

**Summary:**

The paper presents C3B (Comics Cross-Cultural Benchmark), a novel multimodal benchmark specifically designed to evaluate the cultural awareness of Multimodal Large Language Models (MLLMs) through the medium of comics. The work is motivated by the limitations of existing benchmarks, which often rely on real-world images that typically contain only a single cultural context. By using comics, the authors are able to synthesize complex frames containing multiple, overlapping cultural elements, thereby increasing the difficulty of recognition and reasoning. The benchmark consists of over 2000 images and 18000 QA pairs across three progressively difficult tasks: cultural object extraction, cultural-conflict reasoning, and multilingual content generation. This is a timely and important contribution, as cultural sensitivity and cross-lingual capability are increasingly critical for the global deployment of MLLMs. The authors' extensive evaluation of 11 open-source models reveals a significant performance gap compared to humans, underscoring the benchmark's value as a challenging diagnostic tool.

**Reviewer Concerns:**

During the rebuttal phase, the authors successfully addressed several critical concerns raised by the reviewers. The authors provided quantitative evidence using CLIPScore and fine-tuning experiments, demonstrating that their generated comic images effectively encode cultural knowledge that transfers to real-world datasets. The authors provided much-needed details regarding the human-in-the-loop annotation process, including inter-annotator agreement scores, annotator backgrounds, and the correction rate. Concerns regarding grammatical errors and prompt complexity were addressed through a systematic re-evaluation of the prompts, proving that the core experimental conclusions remained robust even after polishing the instructions.

**Reviewer Scores:**

Based on the rebuttal and the subsequent engagement, the final alignment of the reviewers could be as follows:Reviewer h7gx: 2 -> 2. Despite the authors' extensive technical responses and transferability proofs, this reviewer maintained a fundamental philosophical objection to using AI-generated content for cultural benchmarking. However, their concerns regarding "subpar technology" were countered by the authors' empirical evidence.Reviewer oEmH: 2 -> 6. This reviewer was significantly more satisfied after the rebuttal, specifically citing the detailed clarifications on OCR, the addition of an Ethics Statement, and the improved transparency regarding annotator demographics.Reviewer MQS5: 6 -> 8. This reviewer upgraded their score to a strong accept after being convinced by the cross-benchmark evaluation showing that C3B skills transfer to real-world photography.Reviewer 6ZTX: 8 -> 8. This reviewer remained consistently positive, recognizing the innovative use of comics and the logical progression of the multitask structure, even while noting room for future work in non-MT generation tasks.

---

### Decision · Program_Chairs · 2026-01-26

Accept (Poster)